# Physiotherapy in Prehabilitation for Bariatric Surgery—Analysis of Its Impact on Functional Capacity and Original Predictive Models of Functional Status Outcome

**DOI:** 10.3390/jcm14155265

**Published:** 2025-07-25

**Authors:** Katarzyna Gierat-Haponiuk, Piotr Wąż, Julia Haponiuk-Skwarlińska, Maciej Wilczyński, Ireneusz Haponiuk

**Affiliations:** 1Department of Clinical Physiotherapy, Medical University of Gdańsk, 80-210 Gdańsk, Poland; 2Autonomic Physiotherapy Team, University Clinical Center of Gdańsk, 80-219 Gdańsk, Poland; 3Department of Nuclear Medicine, Medical University of Gdańsk, 80-210 Gdańsk, Poland; 4Department of Pediatric Cardiology and General Pediatrics, Doctoral School, Medical University of Warsaw, 02-091 Warsaw, Poland; 5Department of Oncological, Transplant and General Surgery, Medical University of Gdańsk, 80-210 Gdańsk, Poland; maciej.wilczynski@gumed.edu.pl; 6Department of Clinical Physiotherapy, Jedrzej Sniadecki Academy of Physical Education and Sport, 80-336 Gdańsk, Poland; ireneusz_haponiuk@poczta.onet.pl

**Keywords:** prehabilitation, bariatric surgery, physiotherapy, obesity

## Abstract

**Background/Objectives**: Prehabilitation is a multimodal intervention introduced in preparation for various surgical procedures. The most effective treatment for obesity is bariatric surgery. Physiotherapy during prehabilitation for bariatric surgery may be an effective method of functional capacity improvement. We aimed to evaluate the impact of an individual outpatient 12-week, exercise-based physiotherapy program featuring prehabilitation on functional status, exercise tolerance, everyday mobility, and fatigue among patients qualified for bariatric surgery. **Methods**: The completion of an individual outpatient 12-week, exercise-based physiotherapy program during prehabilitation was an inclusion criterion for the study group. Participants included in the study and control groups were assessed twice, after enrollment into the prehabilitation program (the first assessment) and after prehabilitation but before surgery (the second assessment). Both assessments involved functional tests (a six-minute walking test [6MWT], a timed up and go test [TUG], a chest mobility test, anthropometric measures, a mobility index [Barthel], and a modified Borg scale). The collected anthropometric data and values from the 6MWT were used to create original linear models. This study followed STROBE recommendations. **Results**: The study group and control group did not differ statistically in terms of their anthropometric data. Statistically significant results were obtained between the first and second assessments in both groups in terms of body weight and waist circumference. However, only the study group showed improved results in the TUG test (*p* = 0.0001) and distance in the 6MWT (*p* = 0.0005). The study group presented with the normalization of blood pressure (BP) after exertion in the second assessment (systolic BP *p* = 0.0204; diastolic BP *p* = 0.0377), and the 6MWT results were close to the norms. According to the original linear model used to predict performance in the 6MWT, the primary modifiable determinant of exercise tolerance was the participant’s weight, while gender served as a non-modifiable determinant. **Conclusions**: Exercise-based physiotherapy in prehabilitation was associated with improved functional capacity in patients preparing for bariatric surgery, contributing to the improvement in 6MWT results in relation to the norms as well as exercise tolerance. Body weight may be an independent factor determining distance in the 6MWT for patients undergoing prehabilitation for bariatric surgery.

## 1. Introduction

According to the World Health Organization (WHO), obesity and overweight represent a global epidemic, resulting in over 4 million deaths annually. The WHO classifies them as diseases of civilization. In all regions except sub-Saharan Africa and Asia, the population of obese individuals surpasses that of underweight individuals [1]. Obesity is a complex, multifactorial disease that is considered a major public health problem [2]. It is also associated with an increased risk of many non-communicable diseases, including cardiovascular diseases (CVDs); 13 types of cancer; type 2 diabetes mellitus (T2DM); and chronic respiratory diseases, including obstructive sleep apnea (OSA). According to alarming figures from 2022, 60% of adults in Europe are overweight or obese [3].

### 1.1. Treatment Methods for Obesity

Treatment strategies for obesity include both pharmacological and non-pharmacological approaches [4]. According to the American College of Cardiology (ACC), the American Heart Association (AHA), and The Obesity Society (TOS) guidelines, modern obesity treatments include non-pharmacological methods, such as lifestyle changes, and pharmacological and surgical treatment. More specifically, current treatments include nutrition and diet, physical activity, cognitive behavioral therapy, psychological support, pharmacological treatment with anti-obesity medications (AOMs), bariatric and metabolic surgery, and the treatment of comorbidities [5].

There are currently five drugs registered by the European Medicines Agency (EMA) for the treatment of obesity in the European Union (EU): orlistat, a combined formulation of naltrexone hydrochloride and bupropion hydrochloride; liraglutide; semaglutide; and tirzepatide. Although these medications provide specific health benefits solely through weight reduction, they may exert disparate effects on blood pressure, glycemic control, and plasma lipid levels due to their complex mechanisms of action [6].

According to many reports in the literature, surgery remains the most effective treatment for obesity in terms of long-term weight loss, the alleviation of comorbidities, improved quality of life, and reduced overall mortality [7]. The most recent guidelines for surgical treatment, which were developed in 2022 by the American Society for Metabolic and Bariatric Surgery (ASMBS) and the International Federation for the Surgery of Obesity and Metabolic Disorders (IFSO), with the latter including the Society of Polish Surgeons, recommend surgery for patients with a body mass index (BMI) of ≥40 or ≥35 combined with obesity-related comorbidities, including type 2 diabetes. Additionally, bariatric surgery may be considered for patients with a BMI of 30–34.9 kg/m^2^ when conservative management fails to achieve adequate results [8]. The most used surgical methods include sleeve gastrectomy and gastric exclusion with loop or Roux-Y anastomosis [4,6,8].

### 1.2. Prehabilitation in Surgery

The term “prehabilitation” is defined as personalized, multimodal, needs-based interventions to improve an individual’s physiological resilience before an expected major stressor, for example, major surgery [9]. Multimodal prehabilitation includes exercise to improve physical fitness, smoking/alcohol cessation, psychological stress reduction, and nutrition optimization. Thus far, prehabilitation has been studied in specific groups of patients undergoing surgery, mainly in relation to unimodal physiotherapy prehabilitation [10,11]. Physiotherapy in prehabilitation before thoracic and abdominal surgery [12], colorectal cancer surgery [13], and breast cancer surgery [14] has been studied. Based on a review of the literature on a range of surgical conditions, multimodal prehabilitation interventions prior to elective major surgery in adults show encouraging early results, but data on their clinical effectiveness are currently very limited [15].

### 1.3. Prehabilitation in Bariatric Surgery

In 2016, the Enhanced Recovery After Surgery (ERAS) Society published its guidelines [16], which included prehabilitation, or preoperative physical preparation to increase functional capacity and prepare the patient for the metabolic stress associated with surgery. In 2020, the Metabolic and Bariatric Surgery Chapter of the Association of Polish Surgeons (Polish acronym: SCMiB TCHP) published care standards for bariatric and metabolic surgery. These recommendations include numerous consultations and investigations during the preoperative period, including dietary consultation, psychological consultation, and care in the form of internal medicine. However, these guidelines do not include exercise- or physiotherapy-based interventions [17].

Current research indicates a paucity of scientific literature addressing the preparation of patients scheduled for bariatric surgery. There is little information indicating what intervention should be used to improve the vital and functional parameters of patients being prepared for surgery. Garcia-Delgato et al., in 2021, published the protocol of a randomized controlled trial and the results of a pilot study on this topic [18]. Unfortunately, there were major difficulties related to patient adherence, and the results of the study have not yet been published. A literature review performed by Herrera-Sentelies et al. [15] after analyzing only 5 eligible articles out of 4550 showed that supervised exercise can have a positive effect on body composition, functional capacity, and quality of life, but there was insufficient evidence regarding surgical outcomes.

To the best of our knowledge, the effects of physiotherapy-based interventions during prehabilitation for bariatric surgery have not been studied. Additionally, the potential determinants of a patient’s functional status prior to bariatric surgery have not yet been described. In this report, we describe the results of an original study regarding the impact of exercise-based physiotherapy during prehabilitation on the functional status of patients enrolled in the Specialized Care Program for Patients with Obesity who were awaiting bariatric surgery, and we propose original predictive models of functional status outcomes.

### 1.4. Aim and Hypotheses

The aim of this cross-sectional, prospective study was to evaluate the effect of individually tailored, outpatient, 12-week exercise-based physiotherapy on functional status, as assessed by functional tests, exercise tolerance and physical performance, anthropometric measures, everyday mobility, and subjective fatigue rating scales, during the prehabilitation of patients who qualified for bariatric surgery and participated in the Specialized Care Program for Patients with Obesity.

The additional objectives of this study were as follows:-To compare the participants’ functional test outcomes with the Polish Respiratory Society [19] and literature-based normal values [20].-To formulate an original predictive linear model of functional status outcomes among patients awaiting bariatric surgery.


Research hypotheses:


1. The implementation of exercise-based physiotherapy during prehabilitation positively influences the functional status of patients being prepared for bariatric surgery.

2. Implementing an exercise-based physiotherapy program during prehabilitation for bariatric surgery improves exercise tolerance and physical performance.

3. Such an exercise-based physiotherapy program assists in body weight and abdominal circumference reduction.

4. Respiratory exercises during prehabilitation increase chest mobility and reduce the feeling of fatigue while at rest and after exercise in patients awaiting bariatric surgery.

5. Implementing an exercise-based physiotherapy program during prehabilitation normalizes heart rate and blood pressure values among patients awaiting bariatric surgery.

## 2. Materials and Methods

This cross-sectional, single-center, prospective study was launched at the University Clinical Center of Gdansk, Poland, by the Autonomic Physiotherapy Team researchers between 2022 and 2024. The participants were recruited from the Outpatient General Surgery Clinic for Obesity and Metabolic Diseases Treatment Center, the University Clinical Center of Gdansk, Poland, at the time of their enrollment into the Specialized Care Program for Patients with Obesity and qualification for bariatric surgery.

### Specialized Care Program for Patients with Obesity

The Specialized Care Program for Patients with Obesity was a program that provided comprehensive medical care to patients qualified for bariatric surgery. The program was implemented to improve care among patients being surgically treated for obesity and was active from 2022 to 2025. The program involved the following:-The preparation of the patient for bariatric surgery (including health condition counseling, diagnostic tests, and consultations);-Preoperative care, specifically a 3–6-month period of care provided by a multi-specialist team, including surgeons, specialists in internal medicine or diabetology, anesthesiologists, physiotherapists, psychologists, and dietitians;-The final qualification of the patient for bariatric surgery;-Surgery;-A follow-up visit after surgery, to be implemented within seven to 14 days after discharge.


Study group:


Fifty-two consecutive patients were enrolled in the Specialized Care Program for Patients with Obesity and qualified for bariatric surgery within two months of recruitment. Fifty patients met the study inclusion criteria and did not meet the exclusion criteria. Two patients qualified for inpatient rehabilitation at a rehabilitation center due to multimorbidity and were excluded from this study.

The inclusion criteria were as follows:-BMI > 40;-Participation in the Specialized Care Program for Patients with Obesity;-Qualification for bariatric surgery.

The exclusion criteria were as follows:-Disqualification from the Specialized Care Program for Patients with Obesity;-Cancer (history or active);-Significant physical disability or a musculoskeletal disorder at the time of enrollment;-Serious medical conditions, such as myocardial infarction, stroke, acute heart failure, or pulmonary embolism within 30 days prior to study enrollment;-An inability to participate in outpatient physiotherapy.

The enrolled patients were then divided into two groups:-A study group, including 30 patients who were given exercise-based physiotherapy during prehabilitation.-A control group, including 20 patients who were not given exercise-based physiotherapy during prehabilitation.

The 20 patients who refused to participate in the exercise-based physiotherapy program or did not attend a satisfactory number of physiotherapy sessions (more than 40% of meetings) constituted the control group. They mentioned that difficulties involved in traveling to the physiotherapy sessions and their work commitments were the reasons for their refusal. They were informed about the benefits of physical activity and invited to attend a follow-up final examination.


Study protocol:


The participants were assessed twice, the first time immediately (one to two days) after enrollment in the Specialized Care Program for Patients with Obesity (the first assessment) and the second time after prehabilitation for bariatric surgery but before the surgery (i.e., 4–5 months after the first physiotherapy visit (the second assessment)).


Individual participant assessment:


The participants were measured individually by the study team members, who were members of the Autonomic Physiotherapy Team at the University Clinical Center of Gdansk. Each participant was measured by the same two physiotherapists to eliminate any potential bias.

The participants’ assessments (first and second) consisted of the following:

A physical examination, including vital signs and anthropometric measures. This examination included the measurement of body mass, waist circumference, and vital signs before and after exertion. Body mass was measured using a single physical scale and without the patients wearing shoes or heavy clothes. After these measurements, BMI was calculated. Waist circumference was measured with a measuring tape. During the measurement, the patient stood upright with their feet together and took several natural breaths, and the measurement was taken halfway between the lower edge of the last rib arch and the apex of the iliac crest [21]. Vital signs, including heart rate (HR) and blood pressure (BP) saturation, were measured before and after exertion. The same automatic sphygmomanometer was used for the HR and BP measurements, and a pulse oximeter was used to measure saturation.

Functional tests: The six-minute walk test (6MWT) [19], the timed up and go test (TUG test) [22], and chest mobility measurement.

Each participant performed functional tests to allow us to determine their functional status. The tests all took place under the same conditions, in the physiotherapist’s office or in the hospital. The six-minute walk test (6MWT) was performed according to the rules provided [19]. The aim of this test was to cover the maximum possible distance in six minutes while walking on a flat surface and following a standardized protocol [19]. The participant was not coached or stimulated to walk quickly, and they could stop and rest as needed. Blood pressure, heart rate, and oxygen saturation were measured at the beginning and end of the test. Maximal heart rate was measured. Prior to the test, the patient was placed in a sitting position to measure their vital signs while at rest. The anthropometric data collected and the values obtained from the 6MWT were used to create separate linear models for women and men.

Balance and physical performance were assessed using the TUG test. The participant was given instructions prior to the test, and a 3-meter distance was marked. The participant's tasks were as follows: at a signal, the participant was required to stand up, walk the marked distance at their fastest pace, turn 180°, return to the starting line, turn 180° again, and sit down. The time was measured from the issuing of the “start” command until the participant was sitting again [22].

For the chest mobility test, the difference between the participant’s maximum inhaled and exhaled chest circumference was determined. The chest circumference was measured at the level of the axillary fossa with a tailor’s centimeter [23].

2.Assessment of activities of daily living: The Barthel mobility index.

A questionnaire-based test of independence in terms of activities of daily living was carried out using the Barthel index [24].

3.Assessment of fatigue: The Borg Rating of Perceived Exertion (RPE) scale [25].

Participants reported their perceived fatigue and dyspnea using the Borg RPE scale (a 10-point scale) before and after the 6MWT.


Physiotherapy during prehabilitation for bariatric surgery:


Based on the results of the first assessment of each participant, an individualized, exercise-based, 12-week physiotherapy program was developed. The program’s target was a moderate level of fatigue on the Borg RPE scale, specifically a 6 on this 10-point scale. The heart rate during exercise could not exceed the maximum heart rate recorded during the 6MWT. This program was implemented in a hybrid format, partially at home and within a physiotherapy studio during therapy sessions with a physiotherapist, comprising a total of ten group meetings.

During these sessions, adjustments were made to the quality and intensity of the exercises performed by the participants at home. The participants attended the facility one to three times per week, engaging in 30 min of group full-body exercise (general conditioning exercises), followed by 30 min of activities utilizing exercise equipment, which was intended to activate various muscle groups [8]. Participants could attend the facility one to three times a week at their own discretion, but all participants attended ten physiotherapeutic meetings in the facility during the 12-week program.

At home, the participants were provided with a sample routine consisting of six to ten general strengthening exercises, which they were instructed to perform three times per week in three sets of 15 repetitions, along with respiratory exercises consisting of six repetitions per set. Additionally, aerobic activities were suggested based on individual participant capabilities, including Nordic walking, walking, cycling, stationary biking, treadmill walking, and swimming. Initially, the duration of these aerobic sessions was set at 20 min, and this duration was progressively increased to 45–60 min per day, maintaining a fatigue intensity of 6 on the Borg RPE scale. These aerobic exercises were to be completed at least three times per week, excluding days during which the participants took part in facility exercises. Participants were required to monitor their heart rate and blood pressure before and after each exercise session, with a recommendation not to exceed a heart rate of 120 beats per minute or a blood pressure of 160/100 mmHg [26].

The group exercise sessions conducted at the center utilized station-based training protocols and equipment, including gymnastic sticks, Thera-Band resistance bands, balls, and a multi-gym. Aerobic exercises were incorporated via cycling on a cycloergometer or walking on a treadmill, utilizing XRISE CYCLE cardiowise^®^ (Pirmasens, Germany) ERM-200 from the ITAM company in Zabrze, Poland devices. Respiratory exercises were integrated into all activities, with patients’ heart rate during exercise being closely monitored and their blood pressure being measured using an ELEKTRO-MED GERATE^®^ (Medtronik, Meerbusch, Germany) device from the ITAM company in Zabrze, Poland as recommended. Patients were instructed regarding how to perform these exercises safely while maintaining a perceived exertion level of 5–6 on the Borg RPE scale, ensuring their exercise heart rate did not exceed that recorded during the 6MWT [26]. Adverse events were systematically monitored in accordance with the protocols of the treatment center. In the rehabilitation facility, the exercises were controlled by the physiotherapist, who recorded whether the session was held in their medical records [9].

Throughout the Specialized Care Program for Patients with Obesity, the program participants underwent various diagnostic assessments, including abdominal ultrasound, polysomnography, gastroscopy, and blood tests to measure glucose and cholesterol levels. They received consultations with a dietitian regarding proper nutrition and consultations with specialists, such as surgeons, internists, psychiatrists, and anesthesiologists, based on their individual needs. No modifications of patients’ pharmacological blood pressure treatments were noted during the physiotherapy intervention. In our study, we did not note major differences in the comorbidity profiles between participants and those who refused to participate and, thus, constituted the control group.

Bias sources and bias mitigation:

The control group was composed of patients who were referred for bariatric surgery but withdrew from the exercise-based physiotherapy program during the prehabilitation for the surgery. Thus, our results were affected by significant selection bias. All the participants who composed the study group received individualized physiotherapy, which was adjusted to meet their needs. The study group exercised with experienced physiotherapists and attended the facility the same number of times. We used the same assessment methods and standardized protocols with both groups to ensure replicability and reliability.


Safety precautions and ethical concerns:


This study was conducted in accordance with the guidelines of the Helsinki Convention and with the approval of the Independent Bioethics Committee at the Medical University of Gdańsk (No. NKBBN/330/2022). The physiotherapy program was conducted by certified physiotherapists, and no adverse events were reported among the members of the study group.


Statistical analysis:


Statistical analysis was performed using the functions and procedures of the R package version 4.1.0 [27]. For quantitative variables, basic statistics were calculated, including the mean, median, standard deviation, and first and third quartiles. Normally distributed variables were described using means and standard deviations, while non-normally distributed variables were characterized using medians and interquartile ranges (the first and third quartiles). For qualitative variables, the primary statistic was the frequency of occurrence of a specific group in the collected research material.

The Shapiro–Wilk test was used to assess whether the values of quantitative variables were normally distributed. When comparing two samples of a quantitative variable, Student’s *t*-test or the Wilcoxon test was applied, depending on the Shapiro–Wilk test result. The homogeneity of variances of the samples compared was also investigated. For all the statistical tests used, we calculated effect sizes using established procedures and R functions, along with 1000-iteration bootstrap simulations where applicable, and then conducted a comprehensive post hoc power analysis for each test. Fisher’s exact test and the χ^2^ test were used to determine whether there was a significant difference between observed and expected frequencies in the constructed contingency tables. Linear regression models were also developed and optimized using a stepwise algorithm, minimizing the Akaike Information Criterion (AIC) parameter. The significance level was set at α = 0.05 for all applied tests.

## 3. Results

### 3.1. First Assessment—Prior to Prehabilitation

The baseline characteristics of variables such as age, gender, weight, BMI, and abdominal circumference for the study and control groups are presented in Table 1. The tests used did not show statistically significant differences between the groups for any of the variables listed above. None of the baseline anthropometric characteristics differed significantly between the study and control groups (Table 1), indicating that the groups were well matched. Prior to physiotherapy during prehabilitation, the values obtained based on the thoracic mobility test were not statistically different between the groups. The baseline TUG time was significantly faster (shorter) in the control group.

Comparisons of the distributions of trait values (vital signs and Borg RPE scale values) in the study groups (Table 1) allow us to conclude that the groups were well matched. The results of the tests used are not statistically significant. The participants were subjected to the 6MWT.

There were no statistically significant differences regarding the anthropometric measures or the functional tests performed in the second assessment between the study and control groups (Table 2).

The values of the parameters described in Table 3 were then measured again. The results of the tests comparing the values of the study and control groups were not statistically significant.

The control group achieved a statistically significantly higher mean distance value in the 6MWT than the test group (*p* = 0.0399) and a lower average time in the TUG test (*p* = 0.0084). In the first case, the significant statistical result obtained, along with the relatively low power of the test, suggests that the actual effect in the population may be larger than the observed value and that the analysis performed, due to the limited sample size, may not have detected all significant differences between the groups (Table 1).


Second assessment—after prehabilitation but before the bariatric surgery


Prior to the 6MWT test, no statistically significant differences were observed between the distributions of values describing vital signs and the Borg RPE scale between the study and control groups (Table 2). There was a statistically significant difference in mean systolic and diastolic blood pressure after the 6MWT test, with a mean of 128.8 mmHg in the study group and 142.1 mmHg in the control group (*p* = 0.0204). The mean diastolic pressure in the study group was 79.2 mmHg. In the control group, this value was 86.4 mmHg (*p* = 0.0377). The other parameters were not statistically significantly different between the groups (Table 2).

For most variables, the observed differences between the groups were small enough that, given the current sample size and the high variability of the data, the statistical tests did not have sufficient power to detect potential statistical significance. Consequently, the results do not allow us to reject the null hypothesis but do not rule out the existence of small effects in the population (Table 1 and Table 2).


Comparison of the first and second assessment results


In addition to comparative analyses between the study and control groups, we examined how the values of the individual parameters changed between the first and second assessments for each group separately. Statistically significant results were obtained when comparing the values obtained from the first and second assessments in the study group for such parameters such as body weight (*p*-value < 0.0001), waist circumference (*p*-value < 0.0001), the TUG test (*p*-value = 0.0001), and distance measurements in the 6MWT (*p*-value = 0.0005; Figure 1). A statistically non-significant result was obtained only when examining differences in value distributions for the chest mobility test (*p*-value = 0.3338). In the control group, the results were not statistically significant when comparing the first and second assessments for parameters such as the chest mobility test (*p*-value = 0.0922), the TUG test (*p*-value = 0.1705), and the distance covered by patients during the 6MWT (*p*-value = 0.0628). Statistical significance was observed for parameters such as body weight (*p*-value = 0.0002) and waist circumference (*p*-value = 0.0143). In each case, the Wilcoxon signed-rank exact test was used to examine the differences in the distributions of the above parameters. The results regarding the Barthel index did not differ between the groups and were in line with the established norms in both assessments.

### 3.2. Comparison of Results to Norms

Based on the collected anthropometric data for the study patients and using the Polish Lung Association guidelines [19] on the methodology and interpretation of the 6MWT and the formula derived from Enright RL et al. [20], theoretical distances were determined for each patient in the study and control groups. The formulas given in this paper also made it possible to determine the lower limit of normal (LLN) value for each patient (Table 4) [19]. The distance values thus obtained were compared with the values measured in the 6MWT.

The mean LLN values calculated in the first assessment were statistically significantly lower than the measured mean distances for both the study and control groups (*p*-values from the student’s *t*-test: 0.000012 and 0.000017, respectively). Conversely, the mean distances calculated using the 6MWT model (first assessment) were statistically significantly higher than the mean distances achieved by the patients in the study and control groups (*p*-values from the student’s *t*-test: 0.000263 and 0.001159, respectively).

In the second assessment of the study group, the distribution of measured distances from the 6MWT did not show a statistically significant difference from that of the values derived from the 6MWT model (*p*-value for the Wilcoxon test: 0.800470). However, the sum of the ranks assigned to the actual distance values was statistically significantly higher than the sum of the ranks assigned to the model-based LLN values (*p*-value for the Wilcoxon test: 0.00001). For the control group, in the second assessment, the mean measured distance was statistically significantly lower than the mean value calculated from the 6MWT model but higher than the mean LLN value (*p*-values from Student’s *t*-test: 0.030534 and 0.000045, respectively).

### 3.3. Linear Models for 6MWT Results

The collected anthropometric data and the values obtained from the 6MWT were used to create separate linear models for women and men. The dependent variable in these models was the distance achieved by the patient in the 6MWT for the first distance measurement. Age, weight, and height for the combined control and study groups were included as independent variables. To determine the optimal set of independent variables, a stepwise algorithm minimizing the AIC was applied. The obtained model-fitting parameters are presented in Table 5.

The model optimization process for men resulted in reducing the number of independent variables to two (height and weight). Only the coefficient value determined for the weight variable was statistically significant. The determined coefficient values indicate that as weight increased, distance decreased at nearly a 1 m–1 kg ratio. In the optimized model for women, there were also two independent variables. The only statistically significant value was the coefficient for the weight variable. The second independent variable in the women’s model was age, rather than height, as in the men’s model. The coefficient value determined for the weight variable was statistically significant. It showed that as weight increased, the distance traveled decreased. The distance reduction per weight increase was slightly greater than in the case of men.

The model for males explained 27.2% of the variance (R^2^ = 0.272), but after adjusting for the number of predictors, this value dropped to 18.7% (adjusted R^2^ = 0.187). For females, the coefficients were even lower: R^2^ = 0.116 (12% of the explained variance) and adjusted R^2^ = 0.084 (8%).

## 4. Discussion

Our prospective study showed that participation in the individualized, outpatient, 12-week, exercise-based physiotherapy program was associated with improved functional capacity among patients in multi-disciplinary prehabilitation for bariatric surgery. This is the first prospective study using a Polish population of bariatric patients, and the results highlight the importance of exercise-based physiotherapy in the preoperative care process.

Based on the results obtained from the study participants during the initial assessment (the first assessment), an original linear model was developed to predict performance in the 6MWT separately for women and men. In this model, the independent variable for both genders was baseline weight; however, women’s predictors also included age, while men’s included height. This represents the first mathematical model proposed and described for predicting distances achieved in this functional test for patients prior to bariatric surgery. The model acknowledges that the primary modifiable determinant of exercise tolerance is the patient’s weight, while gender serves as a non-modifiable determinant. Additionally, age acts as another non-modifiable factor for women, and height is a non-modifiable factor for men.

Although there has been emerging research indicating the impact of physical activity among obese patients, there is little evidence of the impact of preoperative interventions (prehabilitation), as most studies focus on the postoperative rehabilitation process [28,29,30]. To date, studies focused on interventions involving physical training in the preoperative period have been rare. Marcon RE et al., who investigated a group of 22 patients, demonstrated that a four-month low-intensity physical activity training program resulted in significantly improved functional status, specifically, an increased distance in the 6MWT and a decreased resting heart rate [31]. Additionally, a small cohort trial of six months of high-intensity training demonstrated improvements in VO2 max during exercise testing prior to bariatric surgery [32]. Our study aimed to examine the effects of individually designed exercise-based physiotherapy interventions, when supported with dietary support, psychological assistance, and medical care, on fitness indices, exercise tolerance, and body weight. However, our results cannot directly support the clinical outcomes of bariatric surgery, as we focused on functional status during the prehabilitation period. Future studies analyzing surgical and long-term outcomes are warranted to assess probable improvements.

In our study, the study and control groups were matched based on anthropometric parameters and showed no significant differences in independence, as measured by the Barthel mobility index. In the initial assessment of functional parameters, the control group achieved a shorter time in the TUG test and a longer distance in the 6MWT than the study group. Better fitness levels on the part of the control participants may have contributed to their withdrawal from the outpatient exercise-based physiotherapy program before bariatric surgery. This baseline discrepancy is problematic in terms of attributing improvements to the intervention. The intervention group had more room for improvement, while the control group was relatively fitter initially. Withdrawal from physical activity is often described in similar studies, making it difficult to assemble a statistically significant study group [31,33].

Subsequent comparisons between the study and control groups before and after the implementation of prehabilitation indicated that the participants in the study group, following a 12-week, individualized, outpatient exercise-based physiotherapy program, experienced improvements in exercise tolerance. This was shown by reduced systolic and diastolic blood pressure values, which approached normal levels after the 6MWT, in comparison with the control group, which continued to have elevated BP values. In contrast, other studies of the implementation of exercise during the preoperative period did not show an improvement in blood pressure despite similar objectives, such as a 12-week training program prior to bariatric surgery, and also observed improvements in patients’ functional status [31,34]. We expect that the normalization of blood pressure in the perioperative period may contribute to the reduction in perioperative cardiovascular risk and medication burden in patients referred for bariatric surgery [35], but further analysis of postoperative outcomes is needed to confirm this.

Both groups participating in the coordinated specialist care program achieved significant weight loss and abdominal circumference reduction during prehabilitation. The dietary intervention and multiple specialist consultations were sufficient to reduce weight during prehabilitation before bariatric surgery. In addition, both the study group and the control group showed comparable levels of fatigue on the subjective Borg RPE scale before and after the 6MWT. However, only the group participating in physiotherapy showed improved functional status; specifically, the time in the TUG test was significantly shorter, and the distance in the 6MWT was significantly longer. Previous studies have shown that both 12 weeks of stationary training three times per week (eight patients) and telerehabilitation (12 patients) prior to bariatric surgery resulted in longer distances in the 6MWT as well as higher scores regarding the amount of exercise performed compared with the control group.

Reports on interventions prior to bariatric surgery are very limited. Most studies of exercise in preparation for bariatric surgery have focused on weight reduction and quality of life [34]. There is a lack of consensus on the best methods to use during prehabilitation [28]. There is also a lack of reports regarding the Polish population.

Despite the incorporation of respiratory exercises into the prehabilitation exercise-based physiotherapy program, there were no significant effects on oxygen saturation or chest mobility in the study group. It is important to note that the physiotherapy intervention was characterized by individualized exercises scheduled for patients’ specific needs concerning intensity, the muscle groups to be targeted, and duration. The training protocols documented in the existing literature often fail to account for the individual physical capabilities, functional status, or musculoskeletal limitations of participants in preparation for surgery [31,33,34]. In our study, we implemented an individualized physiotherapeutic intervention strategy in the prehabilitation phase preceding bariatric surgery. The lack of change in oxygen saturation could also have been due to the good initial condition of the study participants, while the lack of change in chest mobility could be due to respiratory exercises that were not ideally adjusted.

In this project, validated functional status assessment tools, such as the 6MWT and TUG test, were employed. Such tools have already been used with patients awaiting bariatric surgery [33,34]. Other tools included an exercise test and ergospirometry [31] as well as polysomnography, which has been evaluated as a highly useful method with which to assess patients before bariatric surgery [28].

The participants’ 6MWT scores were also compared against the norms defined by the Polish Respiratory Society [19] and the lowest acceptable distance (LLN). All measured distance values for both the control group and the study group were higher than the lowest acceptable distance value. At baseline (before prehabilitation, at the first assessment), patients in the study group and the control group achieved statistically significantly shorter distances than the standard determined by the Polish Respiratory Society’s model. After prehabilitation (the second assessment), patients undergoing exercise-based physiotherapy during prehabilitation before bariatric surgery achieved a mean distance close to the norm determined by the Polish Respiratory Society model, in contrast with the control group, for which the mean distance was significantly below the normal value.

In our study, we also examined fatigue and everyday mobility among the patients. However, both the Borg RPE and Barthel index showed no statistically significant changes before and after prehabilitation for both groups. We presume that the patients in our study and control groups were generally not functionally limited; therefore, their mobility and fatigue were normal.

### Limitations of This Study

In this study, we investigated the effects of an exercise-based physiotherapy program on the functional status of patients preparing for bariatric surgery. Of an initial cohort of 50 patients, only 30 engaged in exercise-based physiotherapy, and these served as the study group. The sample size was determined by the number of patients eligible for the program within the study collection timeframe. The sample size was not determined statistically, which did not ensure an initial statistical power for our calculations and comparisons, affecting the generalizability of our results. Although the groups were relatively small, the calculations showed statistically significant differences. Also, previous studies on the topic provided data on much smaller cohorts to support their results [31,34]. The authors plan to publish the results derived from a larger cohort of patients from the Specialized Care Program, including data on their postoperative follow-up.

The control group in our study was not randomly chosen but, rather, self-selected. Withdrawal from physiotherapy, which was the self-selective criterion for the control group, introduced potential selection bias. Also, the control group had better baseline functional test results, which could be a reason for withdrawal and, therefore, the selective bias seen in our results. The activity of the control group was not tracked, and their level of inactivity could not be measured, which could have introduced variability into our results. A randomized controlled trial is needed to confirm our results.

Functional assessment methods, such as the 6MWT and the TUG test, were used. Other validated assessment methods, such as exercise testing and ergospirometry, were not utilized. Furthermore, our analysis did not consider potential obesity-related comorbidities, including cardiovascular diseases, dyslipidemia, and diabetes, which may have further influenced the results obtained. Our results are also based on a Polish cohort of Caucasian patients. Therefore, our results may not be applicable to other races, and further studies are warranted.

Our linear predictive model of the functional status outcome was developed on the data of a small sample size of Caucasian patients referred for bariatric surgery. It should be cautiously tested on a larger cohort of patients to ensure its clinical applicability and validity.

There are few scientific reports on therapeutic options for and the measures affecting the preparation of patients scheduled for bariatric surgery. There is little information indicating what intervention should be used to improve the vital and functional parameters of these patients. There is insufficient information to guide the selection of interventions aimed at enhancing the functional capacity of patients undergoing preoperative preparation before bariatric surgery.

## 5. Conclusions

Exercise-based physiotherapy during prehabilitation was associated with improved functional capacity in patients being prepared for bariatric surgery.Exercise-based physiotherapy during prehabilitation can contribute to the improvement of distance travelled in the 6MWT, including in relation to norms and data in the literature.Exercise-based physiotherapy during prehabilitation can make a significant contribution to improving exercise tolerance in terms of normalizing blood pressure.Body weight may be an independent factor significantly determining distance travelled in the 6MWT for women and men undergoing physiotherapy during prehabilitation before bariatric surgery.

## Figures and Tables

**Figure 1 jcm-14-05265-f001:**
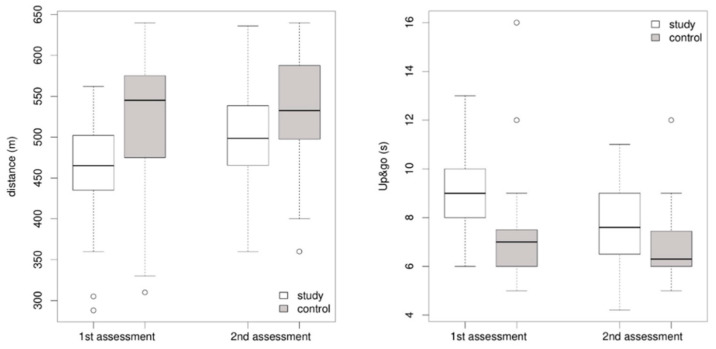
Test results for 6MWT and TUG in the study and control groups in the first and second assessments.

**Table 1 jcm-14-05265-t001:** The characteristics of the study group (subject to physiotherapy intervention) and the control group (not subject to physiotherapy intervention), and the 6MWT results before the test (resting) and after the test (after exertion) (first assessment).

Parameter	Study Group (*n* = 30)	Control Group (*n* = 20)	*p*-Value	Effect Size	Statistical Power
**Anthropometric measures**					
**Age (years)**	43.0 ± 10	40.8 ± 8.2	0.3977 ^t^	0.2367 ^cd^	0.1266
**Gender: female;** **male**	19 11	11 9	0.7683 ^c^	0.0833 ^ph^	0.0906
**Body weight (kg)**	125.5 (105.5; 141.8)	128 (104; 143.5)	0.9763 ^w^	0.0042 ^wr^	0.061 ^mb^
**Height (cm)**	171.5 ± 7.8	174.2 ± 10.1	0.3247 ^w^	0.3036 ^cd^	0.1778
**BMI (kg/m^2^)**	42.3 (37.9; 48.3)	41.7 (37.0; 45.8)	0.4819	0.0995 ^wr^	0.061 ^mb^
**Abdominal circumference (cm)**	132 (118.5; 149)	128.5 (117.0; 138.2)	0.4335 ^w^	0.1108 ^wr^	0.128 ^mb^
**Chest mobility test (cm)**	3.8 (3; 4)	3 (3; 4)	0.9666 ^w^	0.0059 ^wr^	0.040 ^mb^
**TUG (s)**	8.9 ± 1.7	7.5 ± 1.8	0.0084 ^t^	0.8060 ^cd^	0.7811
**Resting parameters**					
**Heart rate (bpm)**	76.3 ± 9.7	77.0 ± 8.8	0.8163 ^t^	0.0662 ^cd^	0.0558
**Systolic pressure (mmHg)**	131.7 ± 14.4	133.3 ± 16.2	0.7283 ^t^	0.1035 ^cd^	0.0643
**Diastolic pressure (mmHg)**	89.4 ± 7.9	89.5 (83.8; 94.2)	0.7358 ^w^	0.0477 ^wr^	0.054 ^mb^
**Saturation (%)**	98.0 (97.0; 98.8)	98.0 (97.0; 99.0)	0.2122 ^w^	0.1764 ^wr^	0.267 ^mb^
**Borg RPE scale (10 degrees)**	0 (0; 0.5)	0 (0; 0.5)	0.5786 ^w^	0.0785 ^wr^	0.075 ^mb^
**Parameters after exertion**					
**Heart rate (bpm)**	113.5 (106.2; 121.8)	119.3 ± 22.4	0.3885 ^w^	0.1220 ^wr^	0.159 ^mb^
**Systolic pressure (mmHg)**	145.0 (130.0; 149.0)	144.5 (132.8; 150.5)	0.7361 ^w^	0.0447 ^wr^	0.065 ^mb^
**Diastolic pressure (mmHg)**	88.4 ± 8.4	89.2 ± 5.9	0.6928 ^t^	0.1070 ^cd^	0.0653
**Saturation (%)**	97.0 (96.0; 98.0)	98.0 (96.8; 99.0)	0.1659 ^w^	0.1960 ^wr^	0.307 ^mb^
**Borg RPE scale (10 degrees)**	2 (0.5; 3)	2 (0.5; 3)	0.6980 ^w^	0.0549 ^wr^	0.079 ^mb^
**Distance (m)**	458.3 ± 65.7	500.5 ± 70.7	0.0399 ^t^	0.6230 ^cd^	0.5616

Basic statistics, for quantitative variables—mean ± standard deviation or median (first quartile; third quartile), ^c^—Pearson’s chi-squared test with Yates’ continuity correction, ^w^—Wilcoxon’s rank-sum test with continuity correction, ^t^—Student’s *t*-test, ^cd^—Cohen’s d, ^wr^—Wilcoxon’s r, ^mb^—bootstrap method with 1000 resamples, and ^ph^—Phi coefficient.

**Table 2 jcm-14-05265-t002:** Characteristics of the study and control group (second assessment).

Parameter	Study Group (*n* = 30)	Control Group (*n* = 20)	*p*-Value	Effect Size	Statistical Power
**Anthropometric measures**					
**Body weight (kg)**	117.5 (101.2; 140.2)	122 (99.4; 137.8)	0.8741 ^w^	0.0224 ^wr^	0.052 ^mb^
**BMI (kg/m^2^)**	39.7 (36.2; 45.1)	39.7 (34.9; 42.9)	0.6416 ^w^	0.0658 ^wr^	0.081 ^mb^
**Abdominal circumference (cm)**	121.5 (109.2;145.0)	121.0 (109.8; 136.0)	0.9367 ^w^		
**Chest mobility test (cm)**	4.0 (3.0; 4.8)	3.2 ± 1.2	0.0874 ^w^	0.2417 ^wr^	0.408 ^mb^
**TUG test (s)**	7.0 (6.0; 7.4)	6.3 (6.0; 7.4)	0.7870 ^w^	0.0382 ^wr^	0.062 ^mb^
**Resting parameters**					
**Heart rate (bpm)**	72.4 ± 10.9	75.8 ± 10.5	0.2744 ^t^	0.3172 ^cd^	0.1898
**Systolic pressure (mmHg)**	120.8 ± 15.3	128.3 ± 17.7	0.1272 ^t^	0.4637 ^cd^	0.3500
**Diastolic pressure (mmHg)**	78.1 ± 9.1	82.7 ± 10.6	0.1231 ^t^	0.4703 ^cd^	0.3584
**Saturation (%)**	98 (97; 99)	98 (97; 99)	0.5027 ^w^	0.0948 ^wr^	0.125 ^mb^
**Borg RPE scale (10 degrees)**	0 (0; 0)	0 (0; 0)	0.9661 ^w^	0.0060 ^wr^	0.041 ^mb^
**Parameters after exertion**					
**Heart rate (bpm)**	115.6 ± 23.4	119.6 ± 19.2	0.5174 ^t^	0.1808 ^cd^	0.0942
**Systolic pressure (mmHg)**	128.8 ± 20.4	142.1 ± 18.2	0.0204 ^t^	0.6782 ^cd^	0.6339
**Diastolic pressure (mmHg)**	79.2 ± 10.5	86.4 ± 12.3	0.0377 ^t^	0.6431 ^cd^	0.5883
**Saturation (%)**	98.0 (96.2; 99.0)	98.5 (97.0; 99.0)	0.6186 ^w^	0.0704 ^wr^	0.063 ^mb^
**Borg RPE scale (10 degrees)**	2 (0.6; 3.8)	2 (0.5; 3.0)	0.6022 ^w^	0.0737 ^wr^	0.083 ^mb^
**Distance (m)**	545.0 (476.2; 573.8)	532.8 ± 79.2	0.7435 ^w^	0.0463 ^wr^	0.078 ^mb^

Basic statistics, for quantitative variables—mean ± standard deviation or median (first quartile; third quartile), ^w^—Wilcoxon’s rank-sum test with continuity correction, ^t^—Student’s *t*-test, ^cd^—Cohen’s d, ^wr^—Wilcoxon’s r, and ^mb^—bootstrap method with 1000 resamples.

**Table 3 jcm-14-05265-t003:** Comparison of parameters across groups for the 1st and 2nd assessments.

Parameter	*p*-Value (Comparison of Study 1 and Study 2 for Study Group (*n* = 30))	Effect Size	Statistical Power	*p*-Value (Comparison of Study 1 and Study 2 for Control Group (*n* = 20))	Effect Size	Statistical Power
Body weight (kg)	<0.0001 ^np^	0.8334	1 ^mb^	0.0003 ^np^	0.8291	1 ^mb^
Abdominal circumference (cm)	<0.0001 ^np^	0.8065	1 ^mb^	0.0143 ^np^	0.5475	0.706 ^mb^
Chest mobility test (cm)	0.3338 ^np^	0.1765	0.145 ^mb^	0.0922 ^np^	0.3863	0.421 ^mb^
TUG test (s)	0.0001 ^np^	0.7045	0.999 ^mb^	0.1705 ^np^	0.3064	0.268 ^mb^
6MWT distance (m)	0.0005 ^np^	0.6384	0.958 ^mb^	0.0628 ^tp^	0.4295	0.446 ^mb^

^np^—Wilcoxon’s signed-rank exact test; ^tp^—Paired *t*-test, ^mb^—bootstrap method with 1000 resamples.

**Table 4 jcm-14-05265-t004:** The mean theoretical distances for the 6MWT and the LLN values for the study group and the control group in the first and second assessments.

Distance Determined from Models	Study Group (1st Assessment)	*p*-Value (For the Values Obtained in the Model with the Actual Distance)	Control Group (1st Assessment)	*p*-Value (For the Values Obtained in the Model with the Actual Distance)
**LLN**	378.5 ± 76.8	0.000012 ^td1^	413.4 ± 66.3	0.000017 ^td1^
**6MWT**	522.7 ± 78.5	0.000263 ^td1^	558.7 ± 67.9	0.001159 ^td1^
	**Study Group** **(2nd Assessment)**		**Control Group** **(2nd Assessment)**	
**LLN**	397.9 ± 72	0.000010 ^wd2^	429.2 ± 67.3	0.000041 ^td2^
**6MWT**	542 ± 73.9	0.800470 ^wd2^	574.5 ± 69	0.030534 ^td2^

^td1^—Student’s *t*-test (values for actual distance travelled are given in Table 1), ^td2^—Student’s *t*-test (values for actual distance traveled are given in Table 2), and ^wd2^—Wilcoxon’s rank-sum test with continuity correction (actual distance values are given in Table 1 and Table 2).

**Table 5 jcm-14-05265-t005:** The height and weight variables in the linear model coefficient for the 6MWT score for men and women.

Coefficient	Estimate	*p*-Value
**Male**		
(Intercept)	−182.2877	0.7185
Weight	−1.1334	0.0258
Height	4.5586	0.1337
Multiple R^2^: 0.2721; adjusted R^2^: 0.1865		
**Female**		
(Intercept)	723.8017	0.00000000601
Age	−1.9185	0.0606
Weight	−1.3952	0.0399
Multiple R^2^: 0.1158; adjusted^2^: 0.08417		

## Data Availability

The data obtained in this study can be shared upon request.

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
