# Peer review of "Physiotherapy in Prehabilitation for Bariatric Surgery—Analysis of Its Impact on Functional Capacity and Original Predictive Models of Functional Status Outcome"

_jcm, 2025, doi:10.3390/jcm14155265_

Round 1
Reviewer 1 Report
Comments and Suggestions for Authors
This manuscript examines the role of an exercise-based physiotherapy prehabilitation program in patients awaiting bariatric surgery, and evaluates its impact on functional capacity (measured by 6-minute walk test, 6MWT; timed up-and-go, TUG; etc.) as well as develops predictive models for functional outcomes. The study is a prospective observational cohort in which 50 patients met inclusion criteria; 30 patients who completed a 12-week supervised physiotherapy program were compared to 20 patients who did not participate (serving as a control group) . Both groups underwent assessments at enrollment (pre-prehabilitation) and after 12 weeks/before surgery (post-prehabilitation) . Outcome measures included anthropometry (weight, BMI, waist), functional tests (6MWT distance, TUG time, chest mobility), a mobility/independence index (Barthel Index), and exertional responses (heart rate, blood pressure, oxygen saturation, Borg rating of perceived exertion).
Major Concerns
- A primary concern is the observational design: patients were not randomized to intervention vs. control, but rather self-selected (or were selected by circumstance) into those who underwent physiotherapy and those who did not. This introduces potential selection bias. Indeed, the control group consisted of patients who refused or could not adhere to the physiotherapy program, often due to travel difficulties or work commitments . Such factors may correlate with underlying differences in health, motivation, or lifestyle that could independently affect outcomes. Notably, the control group had significantly better baseline functional performance than the exercise group (faster TUG and longer 6MWT distance) , despite similar anthropometrics. This suggests the possibility that more functionally able or time-constrained individuals opted out of the exercise program. This baseline discrepancy is problematic for attributing improvements causally to the intervention – the intervention group had more room for improvement, while the control group was relatively fitter initially. The authors should address this in the discussion as a limitation (currently it is not explicitly acknowledged). At minimum, the manuscript should temper causal language, recognizing that improvements in the physiotherapy group vs. stability in controls may partly reflect differences in patient characteristics. Ideally, a randomized controlled trial would be needed to confirm efficacy. While a true RCT may be beyond the scope here, the authors can strengthen confidence in their findings by, for example, comparing change scores between groups (see point #2 below) or performing an adjusted analysis. As-is, the inference that “physiotherapy improves functional capacity” needs to be qualified as an association observed under these conditions, given the non-randomized design.
- The results primarily report within-group changes and separate significance tests for each group (e.g. the study group improved in 6MWT, the control group did not). While this is informative, it would be statistically more robust to directly compare the magnitude of improvement between groups. The authors should consider adding an analysis of the group × time interaction or a direct comparison of change scores. For example, a two-sample test on the pre–post difference in 6MWT distance (or an ANOVA/ANCOVA) would confirm whether the improvement in the study group was significantly greater than that in the control group. As it stands, we must infer the intervention effect indirectly. A formal between-group comparison could strengthen the conclusion that the exercise program led to superior functional gains. If such an analysis was done and is buried in the text, it should be highlighted; if not, it is strongly recommended to include it. Given the baseline imbalance in functional capacity, an ANCOVA adjusting for baseline 6MWT or TUG might be particularly appropriate to quantify the intervention effect. The absence of an explicit between-group comparison is a concern because simply noting significance in one group and not in the other does not prove a statistically significant difference in outcomes across groups.
- The study sample is relatively small (N=50 total, with only 30 in the intervention group and 20 controls). This raises concerns about the statistical power to detect all but large effects. Some results (e.g. 6MWT improvement in the control group with p = 0.0628 ) may have been clinically meaningful yet failed to reach significance due to limited power. The authors do mention that only 30 out of 50 eligible patients ultimately participated in physiotherapy , but they do not discuss how this small sample size (and uneven group sizes) might limit the generalizability of the findings. This should be explicitly acknowledged as a limitation. Moreover, with such a sample, any subgroup or multivariable analyses (such as the predictive linear models) become less reliable (see next point). The manuscript would benefit from a discussion of power – for instance, was a power calculation conducted for the expected improvements in 6MWT or TUG? If not, consider doing a post-hoc power or at least cautioning that smaller effect sizes might not have been detected. Overall, the conclusions should be qualified given the modest sample size. For example, claiming the program “improves functional capacity” should be softened to indicate it “was associated with improvements in this cohort,” unless the statistical evidence is very strong. Encourage the authors to be careful about not over-generalizing beyond this sample.
- The development of an “original predictive model” for 6MWT distance is an interesting aspect of this study, but there are several concerns regarding this analysis. First, splitting the model by sex leaves very small sample sizes for each regression (30 women and 20 men), which can lead to unstable estimates. Indeed, one of the included predictors for men (height) did not reach statistical significance , and for women age was only borderline. The authors should provide additional details to support the validity of these models: for example, report the R² or explained variance, confidence intervals for coefficients, and any model diagnostics. Currently, readers have only the coefficient table (Table 5) and a statement that this is the first such model. Without goodness-of-fit metrics or validation, it’s unclear how well these predictors explain the variance in 6MWT. There is also no external validation – the model is derived and tested on the same small sample, raising the risk of overfitting. The manuscript should caution that these equations are preliminary. Phrasing such as “the first mathematical model proposed…” is acceptable, but the authors must note that further validation is required before clinical use. Additionally, there is a minor labeling issue: in Table 5, the variable “growth” for the men’s model should presumably be “height” – the terminology should be corrected for clarity. Overall, while the attempt at prediction is appreciated, this section needs a more cautious interpretation. Emphasize that weight was found to be the primary modifiable determinant of 6MWT distance (a useful finding) but avoid giving the impression that the provided equations are definitive predictive tools at this stage.
- The authors focus on functional capacity metrics, which is appropriate, but the manuscript would be strengthened by relating these to clinical outcomes if possible. For example, improved 6MWT distance and TUG times are presumed to be beneficial before surgery (e.g. potentially reducing postoperative complications or improving recovery), but the study did not track postoperative outcomes. This is fine given the scope, yet the conclusions could acknowledge this. Currently, the conclusions list improved functional capacity, etc., as benefits of prehabilitation , which is supported by the data, but they do not mention that actual surgical outcomes (length of stay, complications, etc.) were not studied. It would be a valuable discussion point to note that while functional improvements were observed, it remains to be seen if these translate into improved surgical or long-term outcomes – something future studies should assess. Additionally, one of the aims was to look at “fatigue” and “everyday mobility”; however, aside from the Borg RPE (which showed no difference) and the Barthel Index (which was essentially normal for all), there was not much discussion of those aspects. The authors may consider commenting on the lack of change in subjective fatigue levels – perhaps the patients were generally not functionally limited enough at baseline to register a difference in Borg scores. A brief mention in the discussion would show the authors have considered why some measures (e.g. chest mobility, Borg, SpO₂) did not change .
Minor Concerns and Suggestions for Improvement
- The manuscript should clearly describe the physiotherapy program details in one place. It is mentioned that it was an individualized 12-week outpatient program aiming for moderate intensity (Borg ~6/10) , and elsewhere it notes patients attended 1–3 sessions per week (each including 30 min group + 30 min machine exercises) . To avoid confusion, please specify how frequently on average patients exercised and what the typical session entailed in the Methods section. Standardizing this description will help readers reproduce the program. Also, report the adherence more explicitly: how many sessions did participants attend out of those offered (on average)? We know that those who attended <60% of sessions were counted as “control” , but for the 30 who completed the program, a mean attendance rate (and range) would be useful to gauge the “dose” of exercise received.
- The study did not account for patients’ comorbid conditions (e.g. diabetes, cardiac disease, orthopedic issues) in the analysis, as noted by the authors as a limitation . It would be helpful if the authors could provide some baseline description of these factors in the cohort, if available. Even a qualitative statement like “groups had similar distributions of obesity-related comorbidities” or “no major differences in comorbidity profiles were noted between those who opted into physiotherapy and those who did not” would strengthen the reader’s confidence that unmeasured health differences aren’t driving the results. If such data were not collected, then simply acknowledge this and perhaps suggest it be recorded in future studies.
- In Tables 1 and 2, the authors present a thorough comparison of variables. A minor point: labeling could be clarified. For example, in Table 2 (the post-prehab assessment) the header “Baseline characteristics” is a bit confusing when followed by weight, BMI, etc., at the second time point. Perhaps call these “Anthropometric measures” instead of “Baseline characteristics” at 2nd assessment, to avoid misinterpretation. Also, please ensure consistency in decimal formatting (e.g. use either one decimal point or two consistently for mean ± SD). In Table 3 (comparison of 1st vs 2nd within groups) it would help to indicate which test was used for each p-value (though the footnotes np/tp are given). In the figure (Figure 1), make sure it has a proper legend and that axes are labeled clearly (the figure was referenced but not seen in the text excerpt we have – presumably it shows 6MWT and TUG changes). Minor typos: “Parameters afer exertion” in tables should be “after”; “growth” in Table 5 should be “height” .
- Overall the manuscript is understandable, but there are places where the English could be polished for clarity. For instance, in the Results text it reads: “According to the established level of significance, it can be assumed that the distributions of the values of the individual characteristics in Table 1 in each of the study groups belong to the same population.” This could be simplified to “None of the baseline anthropometric characteristics differed significantly between the study and control groups (Table 1), indicating the groups were well matched.” Similarly, the phrase “the TUG test showed a statistically significant shorter mean time in patients in the control group compared to the study group” would be clearer as “baseline TUG time was significantly faster (shorter) in the control group than in the study group.” We encourage the authors to have a thorough read-through or language editing to tighten such phrasing. Another example: the Introduction contains two back-to-back sentences about EMA-approved obesity drugs that are somewhat redundant – consider condensing into one sentence to avoid repetition. Also, be careful with plural vs singular (e.g. “Each participants performed…” should be “Each participant performed…”). These are minor edits that will improve readability.
- The literature review could be slightly expanded regarding previous prehabilitation studies. The authors cite a few small studies (including one with telerehabilitation of 12 patients) , which is good. If there are any recent larger trials or systematic reviews on pre-bariatric exercise (perhaps the one the authors cited that found 5 eligible articles), it would be worth referencing them to highlight how the present study adds new information. Also, when mentioning “lack of reports in the Polish population” , the authors might cite any Polish-specific data or clarify that most data come from other countries. Ensure all references are formatted correctly (the reference list was not fully visible to this reviewer, but numbering in text seems consistent). Check that reference #5 and #6 in the Introduction are not duplicated, as the text around lines 134–143 suggests a repeated citation.
- The finding that the exercise group had significantly lower post-walk blood pressures than controls at the second assessment is intriguing. The authors attribute this to improved exercise tolerance (and indeed it suggests a training effect on cardiovascular response). It might be worth commenting in the discussion that this blunting of exercise-induced blood pressure rise could be beneficial (e.g. indicating improved fitness or lower cardiovascular strain). Also clarify that “normalization of blood pressure after exertion” means that the exercise group’s BP after the 6MWT fell in the normal range, whereas controls remained elevated. Currently the abstract and conclusions mention this normalization, which is great, but one sentence in the discussion interpreting its clinical relevance would be valuable (e.g. possibly reducing perioperative cardiovascular risk, though that’s speculative).
- The use of the Barthel Index showed no differences and essentially all patients were at ceiling (100 points) for independence, as expected for elective surgical candidates. It might be worth explicitly stating in the Results or Discussion that patients were generally fully independent in ADLs, which is why no change was seen in Barthel Index (and that the prehab program was more targeted at exercise tolerance than basic ADLs). This would reassure readers that both groups started with normal daily function, and the intervention was aimed at improving exercise capacity rather than correcting any disability.
Limitations Not Noted by the Authors
The authors have listed several limitations (e.g. lack of cardiopulmonary exercise testing, not accounting for comorbidities) , which we appreciate. However, there are a few additional limitations that were not explicitly mentioned and should be acknowledged:
- Lack of Randomization: As discussed above, the non-randomized nature of group assignment is a significant limitation. This is hinted at (since only 30 of 50 participated in therapy) but not plainly stated as a methodological shortcoming. The authors should explicitly acknowledge that the study design carries a risk of selection bias and confounding, and thus causation cannot be definitively established.
- Baseline Functional Differences: The fact that the control group had better baseline functional test results (TUG, 6MWT) should be mentioned as a limitation in interpreting the results. This difference could confound the comparison of improvements (since the study group started from a lower baseline). The authors did not note this in their limitations section, but it is important to do so, and if possible discuss why it might have occurred (e.g. self-selection of more mobile patients into control).
- Small Sample and Generalizability: While the authors mention the number of participants, they do not explicitly state that the small sample size is a limitation. It is. A total of 50 patients (30 with intervention) from a single center may limit the generalizability of the findings. Results should be confirmed in larger cohorts, and this should be acknowledged.
- Unmeasured Physical Activity in Control Group: The control patients did not attend the supervised program, but it’s not clear whether they engaged in any exercise on their own. They were “informed about the benefits of physical activity” , but their actual activity levels weren’t tracked. This is a limitation because some control patients might have done independent exercise (or conversely, drastically reduced activity), which could influence their outcomes. Future studies might use activity trackers or at least questionnaires to estimate how “inactive” the control group truly was, but in this study we have to assume they maintained usual lifestyle. Acknowledge that this could introduce some variability or dilute differences.
- No Long-Term Outcomes: As noted, the study did not examine whether the improvements in functional capacity led to differences in postoperative outcomes or long-term weight loss maintenance. While this was beyond the scope, it’s worth stating that this remains unknown. The ultimate goal of prehabilitation is to improve surgical results; without data on that, the impact of the functional improvements, though promising, is somewhat indirect. The authors could mention this and possibly propose tracking such outcomes in follow-up research.
Suggestions for Language and Clarity Improvements
- Use more cautious wording given the study design. For example, instead of “Physiotherapy in prehabilitation improves the functional capacity…” , say “appears to improve” or “was associated with improved functional capacity in our cohort.” Small changes like this throughout will ensure the language aligns with the level of evidence.
- Ensure consistency in terminology: Standardize terms like “prehabilitation physiotherapy program” vs “exercise-based physiotherapy program” – choose one and use it consistently. Also, replace non-standard terms (e.g. “growth” -> “height” as noted). Make sure acronyms are defined on first use (6MWT, TUG, etc., which you did define in Abstract/Methods).
- Remove redundant sentences: In the Introduction, combine the two sentences about the five EMA-approved obesity drugs into one, to improve flow . Also check for any repeated ideas across sections and streamline them.
- A careful proofreading is needed. Some sentences are overly long or have structural issues. For instance, “Our study implemented an individualized physiotherapeutic intervention strategy in the prehabilitation phase…” could be split into two for clarity. Ensure subject-verb agreement (“each participant was measured…”, not “were”). Use past tense consistently when describing the experiment and results.
- Clarity in figure/table references: When referring to Figure 1 or tables in the text, explicitly mention what the reader should notice (e.g. “Figure 1 illustrates that 6MWT distance increased in the study group but not in controls”). This will guide readers to interpret the visuals correctly. In the text, you might state the actual values or improvements to complement the figure.
- Double-check that all statements that need a citation have one, and that all references in the list are complete. For example, if you mention guidelines or prevalence data, ensure the reference is up to date. The reference list wasn’t fully available in the review PDF, but ensure formatting follows the journal style (JCM uses numerical brackets, which you have done). Specifically, verify references [31,33,34] which are cited around the discussion of functional assessment tools and limitations – make sure they correspond to appropriate sources (they seem to relate to prior studies in bariatric prehab or assessments).
- The Abstract’s results are somewhat hard to follow with the break in lines . Ensure the final version of the abstract clearly states the key outcomes in a cohesive manner (e.g. include actual numbers or percent improvements if possible, and state that only the exercise group improved in TUG/6MWT). The Conclusion section of the paper currently presents four bullet points – consider writing them as a concise narrative paragraph for the journal (unless JCM prefers bullet conclusions). The bullet points themselves are fine content-wise, but minor rephrasing for clarity could help (e.g. point 2 could say “significantly increases 6MWT distance, approaching normative values for age/sex”).
Author Response
This manuscript examines the role of an exercise-based physiotherapy prehabilitation program in patients awaiting bariatric surgery and evaluates its impact on functional capacity (measured by 6-minute walk test, 6MWT; timed up-and-go, TUG; etc.) as well as develops predictive models for functional outcomes. The study is a prospective observational cohort in which 50 patients met inclusion criteria; 30 patients who completed a 12-week supervised physiotherapy program were compared to 20 patients who did not participate (serving as a control group) . Both groups underwent assessments at enrollment (pre-prehabilitation) and after 12 weeks/before surgery (post-prehabilitation). Outcome measures included anthropometry (weight, BMI, waist), functional tests (6MWT distance, TUG time, chest mobility), a mobility/independence index (Barthel Index), and exertional responses (heart rate, blood pressure, oxygen saturation, Borg rating of perceived exertion).
Major Concerns
- A primary concern is the observational design: patients were not randomized to intervention vs. control, but rather self-selected (or were selected by circumstance) into those who underwent physiotherapy and those who did not. This introduces potential selection bias. Indeed, the control group consisted of patients who refused or could not adhere to the physiotherapy program, often due to travel difficulties or work commitments. Such factors may correlate with underlying differences in health, motivation, or lifestyle that could independently affect outcomes. Notably, the control group had significantly better baseline functional performance than the exercise group (faster TUG and longer 6MWT distance) , despite similar anthropometrics. This suggests the possibility that more functionally able or time-constrained individuals opted out of the exercise program. This baseline discrepancy is problematic for attributing improvements causally to the intervention – the intervention group had more room for improvement, while the control group was relatively fitter initially. The authors should address this in the discussion as a limitation (currently it is not explicitly acknowledged). At minimum, the manuscript should temper causal language, recognizing that improvements in the physiotherapy group vs. stability in controls may partly reflect differences in patient characteristics. Ideally, a randomized controlled trial would be needed to confirm efficacy. While a true RCT may be beyond the scope here, the authors can strengthen confidence in their findings by, for example, comparing change scores between groups (see point #2 below) or performing an adjusted analysis. As-is, the inference that “physiotherapy improves functional capacity” needs to be qualified as an association observed under these conditions, given the non-randomized design.
Thank you for the valuable comment. As you have pointed out this is not a RCT and we have corrected our approach to the conclusions and we have addressed it within the discussion and limitations.
Abstract:
„ Exercise-based physiotherapy in prehabilitation was associated with improved functional capacity in patients preparing for bariatric surgery, contributing to the improvement in 6MWT results in relation to the norms as well as exercise tolerance.”
Page 2 lines 204-207
Discussion:
„ Our prospective study showed that participation in the individualized, outpatient, 12-week, exercise-based physiotherapy program was associated with improved functional capacity among patients in multi-disciplinary prehabilitation for bariatric surgery.”
Page 18, lines 3150-3153
“This baseline discrepancy is problematic in terms of attributing improvements to the intervention. The intervention group had more room for improvement, while the control group was relatively fitter initially.”
Page 19 lines 2247-2249
“The control group in our study was not randomly chosen but, rather, self-selected. Withdrawal from physiotherapy, which was the self-selective criterion for the control group, introduced potential selection bias. Also, the control group had better baseline functional test results, which could be a reason for withdrawal and, therefore, the selective bias seen in our results. The activity of the control group was not tracked, and their level of inactivity could not be measured, which could have introduced variability into our results. A randomized controlled trial is needed to confirm our results.”
Page 21 lines 2420-2427
Conclusions
“1. Exercise-based physiotherapy during prehabilitation was associated with improved functional capacity in patients being prepared for bariatric surgery.
- Exercise-based physiotherapy during prehabilitation can contribute to the improvement of distance travelled on the 6MWT, including in relation to norms and data in the literature.”
Page 23 lines 2661-2666
- The results primarily report within-group changes and separate significance tests for each group (e.g. the study group improved in 6MWT, the control group did not). While this is informative, it would be statistically more robust to directly compare the magnitude of improvement between groups. The authors should consider adding an analysis of the group × time interaction or a direct comparison of change scores. For example, a two-sample test on the pre–post difference in 6MWT distance (or an ANOVA/ANCOVA) would confirm whether the improvement in the study group was significantly greater than that in the control group. As it stands, we must infer the intervention effect indirectly. A formal between-group comparison could strengthen the conclusion that the exercise program led to superior functional gains. If such an analysis was done and is buried in the text, it should be highlighted; if not, it is strongly recommended to include it. Given the baseline imbalance in functional capacity, an ANCOVA adjusting for baseline 6MWT or TUG might be particularly appropriate to quantify the intervention effect. The absence of an explicit between-group comparison is a concern because simply noting significance in one group and not in the other does not prove a statistically significant difference in outcomes across groups.
Thank you for the valuable comment. We did not use ANCOVA because this method is highly controversial. One of the main reasons is that it imposes the constraint that the slope of the regression line between the independent and dependent variables must be identical across all groups. Additionally, ANCOVA can introduce additional bias in observational studies (without randomization). In some cases, ANCOVA removes part of the true effects and also requires a linear relationship between the independent and dependent variables, as well as normally distributed residuals.
The ANOVA test was not used in the analysis because a significant portion of the quantitative data collected in the study groups did not meet the assumptions of this method.
- The study sample is relatively small (N=50 total, with only 30 in the intervention group and 20 controls). This raises concerns about the statistical power to detect all but large effects. Some results (e.g. 6MWT improvement in the control group with p = 0.0628 ) may have been clinically meaningful yet failed to reach significance due to limited power. The authors do mention that only 30 out of 50 eligible patients ultimately participated in physiotherapy , but they do not discuss how this small sample size (and uneven group sizes) might limit the generalizability of the findings. This should be explicitly acknowledged as a limitation. Moreover, with such a sample, any subgroup or multivariable analyses (such as the predictive linear models) become less reliable (see next point). The manuscript would benefit from a discussion of power – for instance, was a power calculation conducted for the expected improvements in 6MWT or TUG? If not, consider doing a post-hoc power or at least cautioning that smaller effect sizes might not have been detected. Overall, the conclusions should be qualified given the modest sample size. For example, claiming the program “improves functional capacity” should be softened to indicate it “was associated with improvements in this cohort,” unless the statistical evidence is very strong. Encourage the authors to be careful about not over-generalizing beyond this sample.
Thank you for the comment. We have addressed the small sample issue as a limitation.
The sample size was determined by the number of patients eligible for the program within the study collection time frames. Although relatively small, the statistical calculations showed statistically significant changes. The authors plan publication of the results on a larger cohort of the patients from the Specialized Care Program with their post-op follow-up.
Effect sizes and power of statistical tests were calculated and are presented in Tables 1, 2 and 3 (page 11-15).
- The development of an “original predictive model” for 6MWT distance is an interesting aspect of this study, but there are several concerns regarding this analysis. First, splitting the model by sex leaves very small sample sizes for each regression (30 women and 20 men), which can lead to unstable estimates. Indeed, one of the included predictors for men (height) did not reach statistical significance , and for women age was only borderline. The authors should provide additional details to support the validity of these models: for example, report the R² or explained variance, confidence intervals for coefficients, and any model diagnostics. Currently, readers have only the coefficient table (Table 5) and a statement that this is the first such model. Without goodness-of-fit metrics or validation, it’s unclear how well these predictors explain the variance in 6MWT. There is also no external validation – the model is derived and tested on the same small sample, raising the risk of overfitting. The manuscript should caution that these equations are preliminary. Phrasing such as “the first mathematical model proposed…” is acceptable, but the authors must note that further validation is required before clinical use. Additionally, there is a minor labeling issue: in Table 5, the variable “growth” for the men’s model should presumably be “height” – the terminology should be corrected for clarity. Overall, while the attempt at prediction is appreciated, this section needs a more cautious interpretation. Emphasize that weight was found to be the primary modifiable determinant of 6MWT distance (a useful finding) but avoid giving the impression that the provided equations are definitive predictive tools at this stage.
Thank you for the comment. The Multiple R² and Adjusted R² values are presented in Table 5 (page 18), along with their appropriate interpretations.
“The model for male explains 27.2% of the variance (R² = 0.272), but after adjusting for the number of predictors, this value drops to 18.7% (adjusted R² = 0.187). For female, the coefficients are even lower: R² = 0.116 (12% of the explained variance) and adjusted R² = 0.084 (8%).
The model for male explains 27.2% of the variance (R² = 0.272), but after adjusting for the number of predictors, this value drops to 18.7% (adjusted R² = 0.187).”
Page 18 lines 2142-2149
- The authors focus on functional capacity metrics, which is appropriate, but the manuscript would be strengthened by relating these to clinical outcomes if possible. For example, improved 6MWT distance and TUG times are presumed to be beneficial before surgery (e.g. potentially reducing postoperative complications or improving recovery), but the study did not track postoperative outcomes. This is fine given the scope, yet the conclusions could acknowledge this. Currently, the conclusions list improved functional capacity, etc., as benefits of prehabilitation , which is supported by the data, but they do not mention that actual surgical outcomes (length of stay, complications, etc.) were not studied. It would be a valuable discussion point to note that while functional improvements were observed, it remains to be seen if these translate into improved surgical or long-term outcomes – something future studies should assess. Additionally, one of the aims was to look at “fatigue” and “everyday mobility”; however, aside from the Borg RPE (which showed no difference) and the Barthel Index (which was essentially normal for all), there was not much discussion of those aspects. The authors may consider commenting on the lack of change in subjective fatigue levels – perhaps the patients were generally not functionally limited enough at baseline to register a difference in Borg scores. A brief mention in the discussion would show the authors have considered why some measures (e.g. chest mobility, Borg, SpO₂) did not change .
We appreciate your insightful remarks concerning manuscript. We would like to explain that the aim of this very paper was to present solely the functional capacity of the prehabilited patients. We did not aim to evaluate the influence of functional capacity on the long-term outcomes: length of stay or post-operative complications. It is a valuable observation and we are planning to make further research on a bigger cohort of patients to consider surgical issues. We have acknowledged lack of clinical and surgical outcomes analysis within the limitations and discussion.
Discussion
“However, our results cannot directly support the clinical outcomes of bariatric surgery, as we have focused on functional status during the prehablilitation period. Future studies analyzing surgical and long-term outcomes are warranted to assess probable improvements.”
Page 19, lines 2236-2240
Discussion – limitations
„Furthermore, our analysis did not consider potential obesity-related comorbidities, including cardiovascular diseases, dyslipidemia, and diabetes, which may have further influenced the results obtained.”
Page 21, lines 2425-2428
Next, Borg scores during a six-minute walking test did not indicate that patients were functionally limited because the patients’ who were qualified for bariatric surgery were within normal range. Therefore this aspect may have affected lack of difference in measures (e.g. chest mobility, Borg, SpO₂) even though the participants did practice during a 12-week program. We have acknowledge it shortly within the discussion.
“Lack of change in oxygen saturation could also have been duebe observed due to initial good initial condition of the study participants, while a lack of change in chest mobility could also be due to notto ideally adjusted respiratory exercises that were not ideally adjusted”
Page 20 lines 2514-2523
Discussion
“In our study, we also examined fatigue and everyday mobility among the patients. However, both the Borg RPE and Barthel index showed no statistically significant changes before and after prehabilitaion for both groups. We presume that the patients in our study and control groups were generally not functionally limited; therefore, their mobility and fatigue were normal.”
Page 21 lines 2396-2401
Minor Concerns and Suggestions for Improvement
- The manuscript should clearly describe the physiotherapy program details in one place. It is mentioned that it was an individualized 12-week outpatient program aiming for moderate intensity (Borg ~6/10) , and elsewhere it notes patients attended 1–3 sessions per week (each including 30 min group + 30 min machine exercises) . To avoid confusion, please specify how frequently on average patients exercised and what the typical session entailed in the Methods section. Standardizing this description will help readers reproduce the program. Also, report the adherence more explicitly: how many sessions did participants attend out of those offered (on average)? We know that those who attended <60% of sessions were counted as “control” , but for the 30 who completed the program, a mean attendance rate (and range) would be useful to gauge the “dose” of exercise received.
Each time participants attended the outpatient program was registered in medical documents so adherence was monitored. However, when patients were exercising at their homes we could only rely on their statement. Participants attended the facility 1 to 3 times per week, engaging in 30 minutes of group exercises followed by 30 minutes of activities utilizing exercise equipment. Participants could attend from 1-3 times weekly at their own discretion, but finally all participants had a record of 10 physiotherapeutic meeting in the facility within 12 week-program.
We have corrected the description of the physiotherapy program within methods.
Methods
“The participants attended the facility one to three times per week, engaging in 30 minutes of group full-body exercise (general conditioning exercises), followed by 30 minutes of activities utilizing exercise equipment, which was intended to activate various muscle groups. (8) Participants could attend the facility one to three times a week at their own discretion, but all participants attended ten physiotherapeutic meetings in the facility during the 12-week program.”
Page 9 lines 1952-1958
- The study did not account for patients’ comorbid conditions (e.g. diabetes, cardiac disease, orthopedic issues) in the analysis, as noted by the authors as a limitation . It would be helpful if the authors could provide some baseline description of these factors in the cohort, if available. Even a qualitative statement like “groups had similar distributions of obesity-related comorbidities” or “no major differences in comorbidity profiles were noted between those who opted into physiotherapy and those who did not” would strengthen the reader’s confidence that unmeasured health differences aren’t driving the results. If such data were not collected, then simply acknowledge this and perhaps suggest it be recorded in future studies.
Thank you for this remark. As we generally did not collect the information about the cormobidities but we know that all of the patients underwent consultations with internal medicine specialists. We did not record any change in blood pressure medications. We have added a statement to the methodology.:
Methods
“In our study, we did not note major differences in the comorbidity profiles between participants and those who refused to participate and thus constituted the control group.”
Page 10 lines 2221-2223
- In Tables 1 and 2, the authors present a thorough comparison of variables. A minor point: labeling could be clarified. For example, in Table 2 (the post-prehab assessment) the header “Baseline characteristics” is a bit confusing when followed by weight, BMI, etc., at the second time point. Perhaps call these “Anthropometric measures” instead of “Baseline characteristics” at 2nd assessment, to avoid misinterpretation. Also, please ensure consistency in decimal formatting (e.g. use either one decimal point or two consistently for mean ± SD). In Table 3 (comparison of 1st vs 2nd within groups) it would help to indicate which test was used for each p-value (though the footnotes np/tp are given). In the figure (Figure 1), make sure it has a proper legend and that axes are labeled clearly (the figure was referenced but not seen in the text excerpt we have – presumably it shows 6MWT and TUG changes). Minor typos: “Parameters afer exertion” in tables should be “after”; “growth” in Table 5 should be “height” .
Thank you for your remarks. We have changed the labelling within the tables to: “Anthropometric measures”, ensured the consistency of notation, delated the typos.
- Overall the manuscript is understandable, but there are places where the English could be polished for clarity. For instance, in the Results text it reads: “According to the established level of significance, it can be assumed that the distributions of the values of the individual characteristics in Table 1 in each of the study groups belong to the same population.” This could be simplified to “None of the baseline anthropometric characteristics differed significantly between the study and control groups (Table 1), indicating the groups were well matched.” Similarly, the phrase “the TUG test showed a statistically significant shorter mean time in patients in the control group compared to the study group” would be clearer as “baseline TUG time was significantly faster (shorter) in the control group than in the study group.” We encourage the authors to have a thorough read-through or language editing to tighten such phrasing. Another example: the Introduction contains two back-to-back sentences about EMA-approved obesity drugs that are somewhat redundant – consider condensing into one sentence to avoid repetition. Also, be careful with plural vs singular (e.g. “Each participants performed…” should be “Each participant performed…”). These are minor edits that will improve readability.
Thank you for your kind remarks concerning the language. We have proofread the manuscript and included your suggestions to clarify the language, correct grammatical mistakes and avoid repetition.
Proof reading
- The literature review could be slightly expanded regarding previous prehabilitation studies. The authors cite a few small studies (including one with telerehabilitation of 12 patients) , which is good. If there are any recent larger trials or systematic reviews on pre-bariatric exercise (perhaps the one the authors cited that found 5 eligible articles), it would be worth referencing them to highlight how the present study adds new information. Also, when mentioning “lack of reports in the Polish population”, the authors might cite any Polish-specific data or clarify that most data come from other countries. Ensure all references are formatted correctly (the reference list was not fully visible to this reviewer, but numbering in text seems consistent). Check that reference #5 and #6 in the Introduction are not duplicated, as the text around lines 134–143 suggests a repeated citation.
Thank you for this remark. We would be delighted to cite large trials or systematic reviews, but unfortunately to our best knowledge, they have not been published on prehabilitation in bariatric surgery. Also, there are no data on Polish population. We have highlighted lack of data on Polish population within the discussion:
Discussion:
“This is the first prospective study using a Polish population of bariatric patients, and the results of highlight the importance of exercise-based physiotherapy in the preoperative care process.”
Page 18 lines 3145-3148
Thank you for the remark regarding the references – we have double checked the references.
- The finding that the exercise group had significantly lower post-walk blood pressures than controls at the second assessment is intriguing. The authors attribute this to improved exercise tolerance (and indeed it suggests a training effect on cardiovascular response). It might be worth commenting in the discussion that this blunting of exercise-induced blood pressure rise could be beneficial (e.g. indicating improved fitness or lower cardiovascular strain). Also clarify that “normalization of blood pressure after exertion” means that the exercise group’s BP after the 6MWT fell in the normal range, whereas controls remained elevated. Currently the abstract and conclusions mention this normalization, which is great, but one sentence in the discussion interpreting its clinical relevance would be valuable (e.g. possibly reducing perioperative cardiovascular risk, though that’s speculative).
Thank you for this kind remark. We added this sentence to “Discussion”:
“This was shown by reduced systolic and diastolic blood pressure values, which approached normal levels after the 6MWT, in comparison to control group, which continued to have elevated BP values.”
Page 19 lines 3249-3252
“We expect that the normalization of blood pressure in the perioperative period may contribute to the reduction of perioperative cardiovascular risk and medication burden in patients referred for bariatric surgery (35), but further analysis of post-operative outcomes is needed to confirm this.”
Page 19 lines 3256-3260
- The use of the Barthel Index showed no differences and essentially all patients were at ceiling (100 points) for independence, as expected for elective surgical candidates. It might be worth explicitly stating in the Results or Discussion that patients were generally fully independent in ADLs, which is why no change was seen in Barthel Index (and that the prehab program was more targeted at exercise tolerance than basic ADLs). This would reassure readers that both groups started with normal daily function, and the intervention was aimed at improving exercise capacity rather than correcting any disability.
Thank you for the remark. When we were recruiting the groups of patients we did not know about the patients’ level of daily function. Therefore we applied the Barthel Index to check it. The inclusion criterion to the study group was that the participant could attend the outpatient physiotherapy activities which was a minimum activity level needed.
We have also added a comment to the discussion about Barthel and Borg scores – please see our response to point 5 above.
Limitations Not Noted by the Authors
The authors have listed several limitations (e.g. lack of cardiopulmonary exercise testing, not accounting for comorbidities) , which we appreciate. However, there are a few additional limitations that were not explicitly mentioned and should be acknowledged:
- Lack of Randomization: As discussed above, the non-randomized nature of group assignment is a significant limitation. This is hinted at (since only 30 of 50 participated in therapy) but not plainly stated as a methodological shortcoming. The authors should explicitly acknowledge that the study design carries a risk of selection bias and confounding, and thus causation cannot be definitively established.
Thank you once again for pointing this out. We have included comment on the randomization within the limitations – please see point 1 of major concerns.
- Baseline Functional Differences: The fact that the control group had better baseline functional test results (TUG, 6MWT) should be mentioned as a limitation in interpreting the results. This difference could confound the comparison of improvements (since the study group started from a lower baseline). The authors did not note this in their limitations section, but it is important to do so, and if possible discuss why it might have occurred (e.g. self-selection of more mobile patients into control).
Thank you for this remark. We have added the information about the selective bias and control group within the limitations.
“Also, the control group hads better baseline functional test results, which could be athe reason forof withdrawal and, therefore, the selective bias seen inof our results.”
Page 21 lines 4414-4416
- Small Sample and Generalizability: While the authors mention the number of participants, they do not explicitly state that the small sample size is a limitation. It is. A total of 50 patients (30 with intervention) from a single centre may limit the generalizability of the findings. Results should be confirmed in larger cohorts, and this should be acknowledged.
Thank you for the comment. We have addressed the small sample issue as a limitation.
“The sample size was determined by the number of patients eligible for the program within the study collection time frames. Although they were relatively small, the statistical calculations showed statistically significant changes. The authors plan to publishcation of the results derived fromon a larger cohort of the patients from the Specialized Care Program, including data on with their post-op follow-up.”
Page 21 Lines 4405-4410
- Unmeasured Physical Activity in Control Group: The control patients did not attend the supervised program, but it’s not clear whether they engaged in any exercise on their own. They were “informed about the benefits of physical activity” , but their actual activity levels weren’t tracked. This is a limitation because some control patients might have done independent exercise (or conversely, drastically reduced activity), which could influence their outcomes. Future studies might use activity trackers or at least questionnaires to estimate how “inactive” the control group truly was, but in this study we have to assume they maintained usual lifestyle. Acknowledge that this could introduce some variability or dilute differences.
Thank you for the comment. We have addressed it as a limitation.
“The activity of the control group was not tracked, and their level of inactivity could not be measured, which could have introduced variability into our results..”
Page 21 lines 4416-4418
- No Long-Term Outcomes: As noted, the study did not examine whether the improvements in functional capacity led to differences in postoperative outcomes or long-term weight loss maintenance. While this was beyond the scope, it’s worth stating that this remains unknown. The ultimate goal of prehabilitation is to improve surgical results; without data on that, the impact of the functional improvements, though promising, is somewhat indirect. The authors could mention this and possibly propose tracking such outcomes in follow-up research.
Thank you very much for this comment. We have added a comment about lack of clinical outcomes analysis into the discussion – please see the response to the point 5 of major concerns.
Comments on the Quality of English Language
Suggestions for Language and Clarity Improvements
- Use more cautious wording given the study design. For example, instead of “Physiotherapy in prehabilitation improves the functional capacity…” , say “appears to improve” or “was associated with improved functional capacity in our cohort.” Small changes like this throughout will ensure the language aligns with the level of evidence.
Thank you for the comment. Please see our response to point 1 major concerns for out correction.
- Ensure consistency in terminology: Standardize terms like “prehabilitation physiotherapy program” vs “exercise-based physiotherapy program” – choose one and use it consistently. Also, replace non-standard terms (e.g. “growth” -> “height” as noted). Make sure acronyms are defined on first use (6MWT, TUG, etc., which you did define in Abstract/Methods).
Thank you for the remark. We have standardized the terms within the manuscript. The manuscript underwent proof-reading with a native speaker.
- Remove redundant sentences: In the Introduction, combine the two sentences about the five EMA-approved obesity drugs into one, to improve flow . Also check for any repeated ideas across sections and streamline them.
Thank you very much for the remark. We have combined the repeated sentence.
- A careful proofreading is needed. Some sentences are overly long or have structural issues. For instance, “Our study implemented an individualized physiotherapeutic intervention strategy in the prehabilitation phase…” could be split into two for clarity. Ensure subject-verb agreement (“each participant was measured…”, not “were”). Use past tense consistently when describing the experiment and results.
Thank you for the remark. The manuscript underwent proofreading.
- Clarity in figure/table references: When referring to Figure 1 or tables in the text, explicitly mention what the reader should notice (e.g. “Figure 1 illustrates that 6MWT distance increased in the study group but not in controls”). This will guide readers to interpret the visuals correctly. In the text, you might state the actual values or improvements to complement the figure.
Thank you for the remark. We believe that our figure is very clear for the readers and the explanation could suggest what the reader should see limiting interpretation and critical assessment of our results.
- Double-check that all statements that need a citation have one, and that all references in the list are complete. For example, if you mention guidelines or prevalence data, ensure the reference is up to date. The reference list wasn’t fully available in the review PDF, but ensure formatting follows the journal style (JCM uses numerical brackets, which you have done). Specifically, verify references [31,33,34] which are cited around the discussion of functional assessment tools and limitations – make sure they correspond to appropriate sources (they seem to relate to prior studies in bariatric prehab or assessments).
Thank you for the remark. We have double-checked the references. It seems the citations are correct, if not, please specify what exactly should be changed about the mentioned citations.
- The Abstract’s results are somewhat hard to follow with the break in lines. Ensure the final version of the abstract clearly states the key outcomes in a cohesive manner (e.g. include actual numbers or percent improvements if possible, and state that only the exercise group improved in TUG/6MWT). The Conclusion section of the paper currently presents four bullet points – consider writing them as a concise narrative paragraph for the journal (unless JCM prefers bullet conclusions). The bullet points themselves are fine content-wise, but minor rephrasing for clarity could help (e.g. point 2 could say “significantly increases 6MWT distance, approaching normative values for age/sex”).
Thank you very much for the comment. We have rephrased the conclusions bullet points according to the previous remarks.
Reviewer 2 Report
Comments and Suggestions for Authors
Dear authors,
Congratulations on your work. The manuscript shows potential for publication; however, I believe it could benefit from several improvements to enhance its methodological quality and overall scientific clarity. Please consider the following points:
Title:
From the title alone, it is evident that the manuscript does not follow the STROBE (Strengthening the Reporting of Observational Studies in Epidemiology) guidelines. I strongly recommend that the authors visit the STROBE website and download the appropriate checklist to restructure the manuscript accordingly. This is a simple yet effective step that can significantly improve the methodological rigor and reduce the risk of bias.
Abstract:
Please include a statement in the methods section of the abstract indicating that the study followed the STROBE recommendations.
Introduction:
The introduction could be better structured. It is recommended to begin with two paragraphs outlining what is already known about the topic. Then, add two additional paragraphs discussing what remains unknown—highlighting gaps in the literature and providing a clear justification for the study. Conclude the section with the study hypothesis, objectives, and potential practical implications. This section should be written in continuous academic prose, without bullet points or itemized lists.
Materials and Methods:
This section would benefit greatly from being revised according to the STROBE checklist, as several essential elements are either unclear or missing. Avoid using subheadings or bulleted formats, as these detract from the scientific tone. Maintaining a standardized structure ensures transparency, reduces bias, and enhances reproducibility by providing sufficient detail for others to replicate the study.
Discussion:
The spacing throughout this section appears inconsistent and could be improved for readability. Additionally, the discussion should better justify the sample loss, which currently lacks sufficient explanation.
Conclusion:
The conclusion should be rewritten as continuous text, avoiding the use of numbering or bullet points. This issue recurs throughout the manuscript and should be addressed to maintain a cohesive academic format.
References:
Please include the DOI for all references where available. This is an important detail that improves accessibility and citation accuracy.
I hope these suggestions contribute positively to the improvement of your manuscript.
Kind regards,
Comments on the Quality of English LanguageThe manuscript is generally understandable; however, there are several areas where the English language could be improved for clarity, coherence, and academic tone. I recommend a thorough review by a fluent English speaker or a professional language editing service to refine grammar, sentence structure, and word choice. Enhancing the language quality will improve the overall readability and impact of the paper.
Author Response
Reviewer 2
Dear authors,
Congratulations on your work. The manuscript shows potential for publication; however, I believe it could benefit from several improvements to enhance its methodological quality and overall scientific clarity. Please consider the following points:
Title:
From the title alone, it is evident that the manuscript does not follow the STROBE (Strengthening the Reporting of Observational Studies in Epidemiology) guidelines. I strongly recommend that the authors visit the STROBE website and download the appropriate checklist to restructure the manuscript accordingly. This is a simple yet effective step that can significantly improve the methodological rigor and reduce the risk of bias.
Thank you for the remark. We did not add the the study design term within the title due to the title characters limit and addition of the models of prediction to the observational study report.
We enclose our strobe checklist to the review response.
Abstract:
Please include a statement in the methods section of the abstract indicating that the study followed the STROBE recommendations.
Thank you for the remark, we have added this information within the abstract.
„The study followed STROBE recommendations.”
Page 1
Introduction:
The introduction could be better structured. It is recommended to begin with two paragraphs outlining what is already known about the topic. Then, add two additional paragraphs discussing what remains unknown—highlighting gaps in the literature and providing a clear justification for the study. Conclude the section with the study hypothesis, objectives, and potential practical implications. This section should be written in continuous academic prose, without bullet points or itemized lists.
Thank you for the comment. We believe our introduction begins with what is known (obesity as a disease, treatment methods for obesity, prehabilitaion in surgery, prehabilitation in bariatric surgery), then what is unknown and it is followed by the aim and hypotheses. We have enclosed a comment regarding the knowledge gap within the introduction.
Introduction
„To the best of our knowledge, the effects of physiotherapy-based interventions during prehabilitation for bariatric surgery have not been studied. Additionally, the potential determinants of a patient’s functional status prior to bariatric surgery have not yet been described.”
Page 4 lines 491-494
Materials and Methods:
This section would benefit greatly from being revised according to the STROBE checklist, as several essential elements are either unclear or missing. Avoid using subheadings or bulleted formats, as these detract from the scientific tone. Maintaining a standardized structure ensures transparency, reduces bias, and enhances reproducibility by providing sufficient detail for others to replicate the study.
Thank you for the remark. We have corrected the description of the excercise-based physiotherapy program as well as we have added „Bias sources and bias mitigation” sub-section following closely the STROBE recommendations.
Methods
“The participants attended the facility one to three times per week, engaging in 30 minutes of group full-body exercise (general conditioning exercises), followed by 30 minutes of activities utilizing exercise equipment, which was intended to activate various muscle groups. (8) Participants could attend the facility one to three times a week at their own discretion, but all participants attended ten physiotherapeutic meetings in the facility during the 12-week program.”
Page 9 lines 1952-1958
.
„Bias sources and bias mitigation
The control group was composed of patients who were referred for bariatric surgery but withdrew from the exercise-based physiotherapy program during the prehabilitaion for the surgery. Thus, our results were affected by significant selection bias. All the participants who composed the study group received individualized physiotherapy, which was adjusted to meet their needs. The study group exercised with experienced physiotherapists and attended the facility the same number of times. We used the same assessment methods and standardized protocols with both groups to ensure replicability and reliability.”
Page 10, lines 2232-2241
Discussion:
The spacing throughout this section appears inconsistent and could be improved for readability. Additionally, the discussion should better justify the sample loss, which currently lacks sufficient explanation.
Thank you very much. The manuscript underwent proof-reading and correction. The spacing between the paragraphs within the discussion highlights discussion on diffenet aspects of the manuscirpt. We have adressed the sample loss reasons within the methods section „study group”. We have adressed the self-selection of the control group within the limitations section of the discussion.
Discussion, limitations
„The sample size was determined by the number of patients eligible for the program within the study collection timeframe. Although they were relatively small, the statistical calculations showed statistically significant changes. The authors plan to publish the results derived from a larger cohort of patients from the Specialized Care Program, including data on their post-op follow-up.
The control group in our study was not randomly chosen but, rather, self-selected. Withdrawal from physiotherapy, which was the self-selective criterion for the control group, introduced potential selection bias.”
Page 21 lines 4414-4424
Conclusion:
The conclusion should be rewritten as continuous text, avoiding the use of numbering or bullet points. This issue recurs throughout the manuscript and should be addressed to maintain a cohesive academic format.
Thank you very much for the comment. We feel that the bullet points highlight the main findings of our work and are accepted by the Journal of Clinical Medicine.
References:
Please include the DOI for all references where available. This is an important detail that improves accessibility and citation accuracy.
Thank you for the remark. DOI is not required by the journal, we can add it however further on if you insist.
I hope these suggestions contribute positively to the improvement of your manuscript.
Kind regards,
Reviewer 3 Report
Comments and Suggestions for Authors
Comments on the manuscript ID: jcm-3715807
Titled: Physiotherapy in prehabilitation for bariatric surgery: analysis of
its impact on functional capacity and original predictive models of
functional status outcome
While the manuscript addresses a clinically relevant and timely question regarding the added value of physiotherapy-based prehabilitation for bariatric surgery candidates, it still exhibits several significant weaknesses and flaws that must be carefully addressed before the work is suitable for publication. These weaknesses include methodological clarity, reporting completeness, statistical accuracy, and interpretive balance (some causal claims are made where only associative evidence exists). Until these issues are addressed through more precise framing, more detailed methodological explanations, and a more cautious interpretation of the results, the paper does not yet meet the criteria expected for acceptance. The remarks below form the basis of my scientific assessment.
- The manuscript employs an unconventional layout, incorporating secondary subtitles within the main sections along with underlined headings and subheadings, a presentation style that is rarely encountered.
- The dual use of a randomized controlled trial and a predictive model analysis without clear differentiation. The simultaneous inclusion of an RCT and a predictive-modeling component obscures the study’s aims and generates confusion around the methods, statistical strategy, and presentation of results.
- The manuscript repeatedly describes itself as a “cross-sectional” study, which is inaccurate. For example, the stated aim reads: “The aim of this cross-sectional, prospective study was to evaluate the effect of an individually tailored, outpatient, 12-week exercise-based physiotherapy on functional status.” Such wording reflects the purpose of a prospective randomized controlled trial, that is, to test a causal effect, not a cross-sectional snapshot.
- Because of the design ambiguity, the manuscript falls short of full adherence to CONSORT standards, which are critical for transparent, high-quality reporting of randomized controlled trials.
- Lines 150–160 present five “research hypotheses” that read more like outcome statements than testable propositions, making the later statistical analyses, reporting, and discussion unnecessarily complex. I suggest consolidating the overlaps and defining one clearly articulated primary hypothesis, supported by one or two secondary hypotheses, to streamline both testing and interpretation.
- The sample size appears insufficient for both the causal comparisons and the multivariable modeling, and no power analysis is reported to demonstrate that the chosen cohort is adequate.
- Random assignments were not used to form study and control groups. Instead, participants who declined or dropped out of the intervention were re-labelled as “controls,” which introduces clear selection bias. These individuals were, in fact, fitter at baseline, as evidenced by their superior TUG and 6MWT scores. Consequently, the so-called control cohort is a self-selected, non-randomized comparison group. Further confusion arises from the statement that some “controls” attended fewer than 60 % of the intervention sessions yet remained classified as controls; the rationale for this categorization needs to be clarified or revised.
- The manuscript does not provide a detailed account of the exercise dosage and its progression, namely frequency, intensity, duration, and modality, making the intervention difficult to replicate.
- Lack of physiotherapy-protocol fidelity: the manuscript does not explain how adherence to the exercise program was tracked, e.g., via session logs or wearable monitors, an omission that is especially problematic given the study’s hybrid (in-person/remote) delivery model.
- Potential confounders are not discussed. It remains unclear whether comorbidities, such as diabetes and hypertension, which can markedly affect functional outcomes, were measured and adjusted for in the analyses.
- Non-parametric tests were correctly substituted when the Shapiro–Wilk test reported non-normality. Yet, the manuscript does not list which variables failed the normality check or whether any α-level adjustment was made for multiple comparisons.
- Effect-size metrics (e.g., Cohen’s d, rank-biserial r, Cliff’s δ) are missing; including them would convey the real-world magnitude of the intervention’s impact, rather than relying solely on p-values.
- Although a predictive model is presented, the manuscript does not explain why a stepwise AIC procedure was selected, nor does it report any over-fitting safeguards, such as shrinkage methods, adjusted R², or bootstrap validation.
- For the BP after 6MWT, the mean change (Δ) should be included rather than only absolute post-values; this highlights the physiological gain.
- Baseline Differences: The control group had better baseline scores on the 6MWT and TUG. How might this have influenced the outcomes?
- The lack of significant change in chest mobility contradicts the hypothesis. Consider discussing possible reasons (e.g., insufficient respiratory exercise intensity).
- Model Limitations: The predictive models for 6MWT are intriguing but rely heavily on weight. Could other variables (e.g., muscle mass, comorbidities) improve predictive power?
- Generalizability: The study population is homogenous (Polish cohort). Would the results apply to diverse populations?
- Tables are information-dense but legible. Check consistency in unit presentation (kg/m² vs kg m⁻²).
- Consider shading alternate rows or bolding statistically significant results for easier scanning.
- Figure 1 (6MWT/TUG results) could benefit from error bars or confidence intervals to visualize variability.
- Watch tense consistency; methods = past tense, discussion = present/past perfect.

The manuscript would also benefit from a thorough language edit; improving the English expression and syntax will markedly enhance clarity and readability for an international audience.
Author Response
Comments on the manuscript ID: jcm-3715807
Titled: Physiotherapy in prehabilitation for bariatric surgery: analysis of
its impact on functional capacity and original predictive models of
functional status outcome
While the manuscript addresses a clinically relevant and timely question regarding the added value of physiotherapy-based prehabilitation for bariatric surgery candidates, it still exhibits several significant weaknesses and flaws that must be carefully addressed before the work is suitable for publication. These weaknesses include methodological clarity, reporting completeness, statistical accuracy, and interpretive balance (some causal claims are made where only associative evidence exists). Until these issues are addressed through more precise framing, more detailed methodological explanations, and a more cautious interpretation of the results, the paper does not yet meet the criteria expected for acceptance. The remarks below form the basis of my scientific assessment.
Thank you for the general comment. In our corrected version of the manuscript we address the issue of a non randomised trial and we have ensured the language aligns with the level of evidence.
Abstract:
„ Exercise-based physiotherapy in prehabilitation was associated with improved functional capacity in patients preparing for bariatric surgery, contributing to the improvement in 6MWT results in relation to the norms as well as exercise tolerance.”
Page 2 lines 204-207
Discussion:
„ Our prospective study showed that participation in the individualized, outpatient, 12-week, exercise-based physiotherapy program was associated with improved functional capacity among patients in multi-disciplinary prehabilitation for bariatric surgery.”
Page 18, lines 2150-2153
“This baseline discrepancy is problematic in terms of attributing improvements to the intervention. The intervention group had more room for improvement, while the control group was relatively fitter initially.”
Page 19 lines 2247-2249
“The control group in our study was not randomly chosen but, rather, self-selected. Withdrawal from physiotherapy, which was the self-selective criterion for the control group, introduced potential selection bias. Also, the control group had better baseline functional test results, which could be a reason for withdrawal and, therefore, the selective bias seen in our results. The activity of the control group was not tracked, and their level of inactivity could not be measured, which could have introduced variability into our results. A randomized controlled trial is needed to confirm our results.”
Page 21 lines 2420-2427
Conclusions
“1. Exercise-based physiotherapy during prehabilitation was associated with improved functional capacity in patients being prepared for bariatric surgery.
- Exercise-based physiotherapy during prehabilitation can contribute to the improvement of distance travelled on the 6MWT, including in relation to norms and data in the literature.”
Page 23 lines 2661-2666
- The manuscript employs an unconventional layout, incorporating secondary subtitles within the main sections along with underlined headings and subheadings, a presentation style that is rarely encountered.
Thank you for the remark. We believe the subheadings add clarity that allows for better understanding of our study design and results.
- The dual use of a randomized controlled trial and a predictive model analysis without clear differentiation. The simultaneous inclusion of an RCT and a predictive-modeling component obscures the study’s aims and generates confusion around the methods, statistical strategy, and presentation of results.
Thank you for the comment. Our study is a observational, cross-sectional study but due to the self-selection of the control group it is not a randomised control trial. We have addressed this limitation within the limitations of the discussion.
Discussion
“The control group in our study was not randomly chosen but, rather, self-selected. Withdrawal from physiotherapy, which was the self-selective criterion for the control group, introduced potential selection bias. Also, the control group had better baseline functional test results, which could be a reason for withdrawal and, therefore, the selective bias seen in our results. The activity of the control group was not tracked, and their level of inactivity could not be measured, which could have introduced variability into our results. A randomized controlled trial is needed to confirm our results.”
Page 21 lines 4421-4429
Our manuscript reports on both the results of the observational study as well as the results of the analysis on the statistical model of prediction of the functional status outcome. Both the results on the effect of exercise-based physiotherapy within prehabilitation for bariatric surgery as well as the probable determining factors of the functional status basing on statistical models have not been published yet and may significantly contribute to the level of knowledge in the topic.
- The manuscript repeatedly describes itself as a “cross-sectional” study, which is inaccurate. For example, the stated aim reads: “The aim of this cross-sectional, prospective study was to evaluate the effect of an individually tailored, outpatient, 12-week exercise-based physiotherapy on functional status.” Such wording reflects the purpose of a prospective randomized controlled trial, that is, to test a causal effect, not a cross-sectional snapshot.
Thank you very much for the comment. Although the wording may suggest that we report randomized controlled trial we later explain witing the methods and discussion section that it is not a randomised controlled trial due to the control group self-selection.
„Bias sources and bias mitigation
The control group was composed of patients who were referred for bariatric surgery but withdrew from the exercise-based physiotherapy program during the prehabilitaion for the surgery. Thus, our results were affected by significant selection bias.”
Page 10, lines 2232-2235
Discussion
“The control group in our study was not randomly chosen but, rather, self-selected. Withdrawal from physiotherapy, which was the self-selective criterion for the control group, introduced potential selection bias. Also, the control group had better baseline functional test results, which could be a reason for withdrawal and, therefore, the selective bias seen in our results. The activity of the control group was not tracked, and their level of inactivity could not be measured, which could have introduced variability into our results. A randomized controlled trial is needed to confirm our results.”
Page 21 lines 4421-4429
- Because of the design ambiguity, the manuscript falls short of full adherence to CONSORT standards, which are critical for transparent, high-quality reporting of randomized controlled trials.
Thank you for the remark. Once again, we did not intend to report a randomised controlled trial. We have followed STROBE recommendations, and we add the strobe checklist to the reviewer response.
- Lines 150–160 present five “research hypotheses” that read more like outcome statements than testable propositions, making the later statistical analyses, reporting, and discussion unnecessarily complex. I suggest consolidating the overlaps and defining one clearly articulated primary hypothesis, supported by one or two secondary hypotheses, to streamline both testing and interpretation.
Thank you for the comment. We believe that all of the five hypotheses were tested by our study and allow to clearly differentiate what has been investigated.
- The sample size appears insufficient for both the causal comparisons and the multivariable modeling, and no power analysis is reported to demonstrate that the chosen cohort is adequate.
Thank you for the remark. We address sample size issue within the limitations section of the discussion.
“Limitations of the study
In this study, we investigated the effects of an exercise-based physiotherapy program on the functional status of patients preparing for bariatric surgery. Out of an initial cohort of 50 patients, only 30 engaged in exercise-based physiotherapy, and thesewhich served as the study group. The sample size was determined by the number of patients eligible for the program within the study collection time frames. Although they were relatively small, the statistical calculations showed statistically significant changes. The authors plan to publishcation of the results derived fromon a larger cohort of the patients from the Specialized Care Program, including data on with their post-op follow-up.”
Page 21 lines 4411-4420
- Random assignments were not used to form study and control groups. Instead, participants who declined or dropped out of the intervention were re-labelled as “controls,” which introduces clear selection bias. These individuals were, in fact, fitter at baseline, as evidenced by their superior TUG and 6MWT scores. Consequently, the so-called control cohort is a self-selected, non-randomized comparison group. Further confusion arises from the statement that some “controls” attended fewer than 60 % of the intervention sessions yet remained classified as controls; the rationale for this categorization needs to be clarified or revised.
Thank you for the remark. The control group issues have been addressed within the limitations of the discussion. Please see response to point 3 above.
- The manuscript does not provide a detailed account of the exercise dosage and its progression, namely frequency, intensity, duration, and modality, making the intervention difficult to replicate.
Thank you for the remark. In our opinion the part “Physiotherapy and rehabilitation for bariatric surgery” describes the exercises well. Our patients could choose Nordic working, cycling and other aerobic exercise at home. In the rehabilitation centre the exercises activated variable muscle groups and were individually tailored to the patient – therfore full identification of all characteristics of the excercises is not possible. However, following your remarks, we have provided more specific information.
“The group exercise sessions conducted at the center utilized station-based training protocols and equipment, including gymnastic sticks, Thera-Band resistance bands, balls, and a multi-gym. Aerobic exercises were incorporated via cycling on a cycloergometer or walking on a treadmill walking, specifically utilizing XRISE CYCLE cardiowise® (Germany) devices.”
Page 9 lines 1975 -1979
- Lack of physiotherapy-protocol fidelity: the manuscript does not explain how adherence to the exercise program was tracked, e.g., via session logs or wearable monitors, an omission that is especially problematic given the study’s hybrid (in-person/remote) delivery model.
Thank you for your remark, it is true we were not tracking the workout carried out by participants at their homes (general strengthening exercise and respiratory exercises and aerobic exercises like walking or cycling), as we did not implement any physiotherapy-protocol fidelity systems, using telemetric devices like wearable monitors or sessions logs. The patients confirmed verbally that they had been exercising. We accepted their statements in good faith and their engagement is visible in our results – the study group improved their functional capacity.
Still, in the rehabilitation facility the exercises were controlled by the physiotherapist who recorded in medical records if the session was held or not. Each session in the facility had two parts: general conditioning (full-body) exercises with station training protocols and equipment including gymnastic sticks, Thera-Band resistance bands, balls, altas and with equipment, and aerobic exercises on a cyclometer, treadmill or other devices.
- Potential confounders are not discussed. It remains unclear whether comorbidities, such as diabetes and hypertension, which can markedly affect functional outcomes, were measured and adjusted for in the analyses.
Thank you for this remark. As we generally did not collect the information about the cormobidities but we know that all of the patients underwent consultations with internal medicine specialists. We did not record any change in blood pressure medications. We have added a statement to the methodology.:
“In our study, we did not note major differences in the comorbidity profiles between participants and those who refused to participate and thus constituted the control group.”
Page 10 lines 2221-2223
- Non-parametric tests were correctly substituted when the Shapiro–Wilk test reported non-normality. Yet, the manuscript does not list which variables failed the normality check or whether any α-level adjustment was made for multiple comparisons.
The selection of baseline characteristics and methods of intergroup comparisons was determined by the results of the Shapiro-Wilk test. Accordingly, it is possible to indirectly determine which quantitative variables come from a population with a normal distribution (in the tables presented, the sample mean is given as a measure of central tendency, and the standard deviation as a measure of dispersion) and which do not have a normal distribution (here, central tendency is described by the median, and dispersion is characterized by the first and third quartiles).
For greater clarity, the description in Statistical analysis has been supplemented.
„Statistical analysis:
Statistical analysis was performed using the functions and procedures of the R package (27). For quantitative variables, basic statistics were calculated, including the mean, median, standard deviation, and first and third quartiles. Normally distributed variables were described using mean and standard deviation, while non-normally distributed variables were characterized using median and interquartile range (first and third quartiles). For qualitative variables, the primary statistic was the frequency of occurrence of a specific group in the collected research material.
The Shapiro–Wilk test was used to assess whether the values of quantitative variables were normally distributed. When comparing two samples of a quantitative variable, a Student’s t-test or the Wilcoxon test was applied, depending on the Shapiro–Wilk test result. The homogeneity of variances of the samples compared was also investigated. For all the statistical tests used, we calculated effect sizes using established procedures and R functions, along with 1,000-iteration bootstrap simulations where applicable, and then conducted a comprehensive post-hoc power analysis for each test. Fisher’s exact test and the χ² test were used to determine whether there was a significant difference between observed and expected frequencies in the constructed contingency tables. Linear regression models were also developed and optimized using a stepwise algorithm, minimizing the Akaike Information Criterion (AIC) parameter. The significance level was set at α = 0.05 for all applied tests.”
Page 10, lines 2250-2576
- Effect-size metrics (e.g., Cohen’s d, rank-biserial r, Cliff’s δ) are missing; including them would convey the real-world magnitude of the intervention’s impact, rather than relying solely on p-values.
Effect sizes and power of statistical tests were calculated and are presented in Tables 1, 2 and 3 (page 11-15).
In Table 4 (page 16), we omitted power and effect size calculations because the values for one of the groups were generated by a statistical model, which makes it impossible to reliably estimate these parameters. Standard computational methods require empirical input for both groups being compared, whereas in this case, one group represents simulated value.
- Although a predictive model is presented, the manuscript does not explain why a stepwise AIC procedure was selected, nor does it report any over-fitting safeguards, such as shrinkage methods, adjusted R², or bootstrap validation.
The Multiple R2 and Adjusted R2 values are given in Table 5 and the appropriate interpretations are given.
“The model for male explains 27.2% of the variance (R² = 0.272), but after adjusting for the number of predictors, this value drops to 18.7% (adjusted R² = 0.187). For female, the coefficients are even lower: R² = 0.116 (12% of the explained variance) and adjusted R² = 0.084 (8%).
The model for male explains 27.2% of the variance (R² = 0.272), but after adjusting for the number of predictors, this value drops to 18.7% (adjusted R² = 0.187).”
Page 18 lines 2142-2149
- For the BP after 6MWT, the mean change (Δ) should be included rather than only absolute post-values; this highlights the physiological gain.
Comparison of differences added to Table 3 (page 11).
- Baseline Differences: The control group had better baseline scores on the 6MWT and TUG. How might this have influenced the outcomes?
Thank you for the remark. We have added the comment about the control group into the limitations of the discussion.
“The control group in our study was not randomly chosen but, rather, self-selected. Withdrawal from physiotherapy, which was the self-selective criterion for the control group, introduced potential selection bias. Also, the control group had better baseline functional test results, which could be a reason for withdrawal and, therefore, the selective bias seen in our results. The activity of the control group was not tracked, and their level of inactivity could not be measured, which could have introduced variability into our results. A randomized controlled trial is needed to confirm our results.”
Page 21 lines 4419-4427
- The lack of significant change in chest mobility contradicts the hypothesis. Consider discussing possible reasons (e.g., insufficient respiratory exercise intensity).
Thank you for the remark. We added a comment about the chest mobility in the discussion.
“The lack of change in oxygen saturation could also have been due to good initial condition of the study participants, while a lack of change in chest mobility could be due to respiratory exercises that were not ideally adjusted.”
Page 20 lines 3315-3319
- Model Limitations: The predictive models for 6MWT are intriguing but rely heavily on weight. Could other variables (e.g., muscle mass, comorbidities) improve predictive power?
Thank you very much for the valuable comment. Probably muscle mass and comorbidities could improve predictive power, but we have not collected these data in our study. Future studies may examine its impact and correct our model assumptions.
- Generalizability: The study population is homogenous (Polish cohort). Would the results apply to diverse populations?
Thank you very much for the remark. Indeed, our results are based on homogenous cohort of Caucasian race. Our results should be checked on different races. We have added a comment on the homogenous cohort in the limitations of the discussion.
Our results are also based on Polish cohort of Caucasian patients. Therefore our results could not be applicable to other races and further studies are warranted.
- Tables are information-dense but legible. Check consistency in unit presentation (kg/m² vs kg m⁻²).
Thank you very much for the comment. The changes have been made.
- Consider shading alternate rows or bolding statistically significant results for easier scanning.
Thank you for the remark. We have shaded alterate rows within the tables.
- Figure 1 (6MWT/TUG results) could benefit from error bars or confidence intervals to visualize variability.
Thank you for the comment. The box plot itself is a visualization of the variability of the data, as it displays the median, interquartile range (box), and whiskers (representing minimum/maximum or multiples of the IQR). Therefore, adding traditional error bars would be an unnecessary element that could make the presented figure unreadable.
- Watch tense consistency; methods = past tense, discussion = present/past perfect.
Thank you for your remark. The manuscript underwent proof-reading.
We look forward to hearing from you in due time regarding our submission and to respond to any further questions and comments you may have.
Sincerely, Authors
Round 2
Reviewer 1 Report
Comments and Suggestions for Authors
Critical Weaknesses Requiring Attention
1. Statistical Analysis Gap: Failure to provide between-group comparisons remains a major methodological flaw
2. Predictive Model Overselling: Insufficient caution about model validity and clinical applicability
3. Incomplete Power Discussion: Limited exploration of generalizability implications
Author Response
Thank you for giving us the opportunity to submit once again a revised draft of the manuscript “Physiotherapy in prehabilitation for bariatric surgery – analysis of its impact on functional capacity and original predictive models of functional status outcome.” for publication in the Journal Clinical Medicine. We appreciate the time and effort that you and the reviewers dedicated to providing feedback on our manuscript. We are grateful for the insightful comments on and valuable improvements to our paper.
We have incorporated most of the suggestions made by the reviewers. Those changes are highlighted
within the manuscript. Please see below, in red, for a point-by-point response to the reviewers’
comments and concerns. All page numbers refer to the revised manuscript file with tracked changes.
Reviewer 1
Critical Weaknesses Requiring Attention
1. Statistical Analysis Gap: Failure to provide between-group comparisons remains a major methodological flaw.
Thank you for your comment.
Our manuscript contains comparison between the study and control group in their antropometric measures and vital signs as well as several tests both groups underwent: chest mobility tests, timed up&go test, Borg RPE scale and Barthel index. After the revisions we have provided effect size as well as statistical power with bootstrap to our statistical comparisons. We believe our study contains sufficient statistical data to suport our findings. If there is insufficient statistical data please specify what could be enhanced to improve our research, excluding ANOVA/ANCOVA (we have justified not using it in the first round of revisions).
Predictive Model Overselling: Insufficient caution about model validity and clinical applicability
Thank you for a valuable comment.
We have added a comment about this limitation of the predictive model to the limitations of the discussion.:
„Our linear predictive model of the functional status outcome have been developed on the data of the small sample size of Caucasian patients referred for the bariatric surgery. It should be cautiously tested on a larger cohort of patients to ensure its clinical applicability and validity.”
Page 22, lines 760-764
Incomplete Power Discussion: Limited exploration of generalizability implications
Thank you for a valuable comment. We have extended the exploration of the power within the limitations section of our discussion.
„The sample size has not been determined statistically not ensuring an initial statistical power to our calculations and comparisons affecting generalizability of our results. Although the groups were relatively small, the calculations showed statistically significant differences. Also, previous studies in the topic provided data on much smaller cohorts to support their results (31,34).”
Page 21, lines 738-743
We look forward to hearing from you in due time regarding our submission and to respond to any further questions and comments you may have.
Sincerely, Authors
Reviewer 3 Report
Comments and Suggestions for Authors
I sincerely thank the authors for their thoughtful responses and the thorough revisions made to the manuscript. These changes have significantly enhanced the clarity and overall quality of the work.
Author Response
Thank you for your valuable time. We also believe in a significat upgrade of our work.
We look forward to hearing from you in due time regarding our submission and to respond to any further questions and comments you may have.
Sincerely, Authors